# Beyond nothingness in the formation and functional relevance of voids in polymer films

Falon C. Kalutantirige [1], Jinlong He[2], Lehan Yao[3], Stephen Cotty[4], Shan Zhou[3], John W. Smith[3], Emad Tajkhorshid[5,6], Charles M. Schroeder[3,4,7,8], Jeffrey S. Moore[1,3,8], Hyosung An[9], Xiao Su[4], Ying Li[2] ✉ & Qian Chen[1,3,4,7,8] ✉

Voids—the nothingness—broadly exist within nanomaterials and impact properties ranging from catalysis to mechanical response. However, understanding nanovoids is challenging due to lack of imaging methods with the needed penetration depth and spatial resolution. Here, we integrate electron tomography, morphometry, graph theory and coarse-grained molecular dynamics simulation to study the formation of interconnected nanovoids in polymer films and their impacts on permeance and nanomechanical behaviour. Using polyamide membranes for molecular separation as a representative system, three-dimensional electron tomography at nanometre resolution reveals nanovoid formation from coalescence of oligomers, supported by coarse-grained molecular dynamics simulations. Void analysis provides otherwise inaccessible inputs for accurate fittings of methanol permeance for polyamide membranes. Three-dimensional structural graphs accounting for the tortuous nanovoids within, measure higher apparent moduli with polyamide membranes of higher graph rigidity. Our study elucidates the significance of nanovoids beyond the nothingness, impacting the synthesis–morphology–function relationships of complex nanomaterials.

Voids—internal empty regions or the nothingness—are ubiquitous in materials across length scales and may reveal the formation mechanisms or affect the function of materials. Our Universe presents the largest example of void space—dense fibres of matter enclosing cosmic voids, which control galaxy formation and structure[1]. Voids play an important role at the opposite end of size too; for example, generic interstitial voids exist even for the close packing of atoms in crystalline solids whilst additional voids can be nucleated and grown upon mechanical stress and are responsible for ductile failure of crystals[2]. At the nanoscale, materials with nanovoids are found particularly useful in catalysis[3], energy[4], separation[5] and medical applications[6]. Nanovoids can provide inner surface areas, molecular or charge transport paths, structural or compositional compartmentalization and alternative mechanisms for mechanical response, giving rise to material

[1]Department of Chemistry, University of Illinois, Urbana, IL 61801, USA. [2]Department of Mechanical Engineering, University of Wisconsin-Madison, Madison, WI 53706, USA. [3]Department of Materials Science and Engineering, University of Illinois, Urbana, IL 61801, USA. [4]Department of Chemical and Biomolecular Engineering, University of Illinois, Urbana, IL 61801, USA. [5]Department of Biochemistry, University of Illinois, Urbana, IL 61801, USA. [6]NIH Resource for Macromolecular Modelling and Visualization, Beckman Institute for Advanced Science and Technology, University of Illinois, Urbana, IL 61801, USA. [7]Materials Research Laboratory, University of Illinois, Urbana, IL 61801, USA. [8]Beckman Institute for Advanced Science and Technology, University of Illinois, Urbana, IL 61801, USA. [9]Department of Petrochemical Materials Engineering, Chonnam National University, Yeosu, Jeollanam-do 59631, South Korea. ✉ e-mail: yli2562@wisc.edu; qchen20@illinois.edu

properties dependent on parameters such as void surface area, void volume and material to void ratio[7,8].

However, despite developments of three-dimensional (3D) imaging methods to characterise macroscopic or micron-sized voids based on optical, ultrasound and magnetic resonance imaging[9,10], unravelling the morphology–function relationships in materials with inner nanovoids has been challenging due to a lack of methodologies to image, quantify and understand irregular structures with the needed nanometre (nm) resolution and penetration depth. State-of-the-art X-ray tomography-based microscopy methods are greatly successful in applications such as imaging 3D void-like defects in electrode materials, but with a limited resolution of tens of nm[11]. In the example of polyamide (PA) membranes, which have intricate, irregular 3D crumples (i.e. protruding nodules) enclosing nanovoids[12,13], transmission electron microscopy (TEM) based tomography was recently used to resolve their 3D morphology with nm resolution[12,14,15]. These PA membranes are used as the active layer in thin-film composites for water reclamation, molecular separation and organic solvent nanofiltration[16–18]. Our group previously built on those efforts and developed morphometry methods to elucidate a quantitative relationship among synthesis conditions, morphology, mechanical properties and separation performance of PA membranes[13,19], but only on simple PA membranes, where 3D crumples (and the enclosed voids) are spatially separated. The PA membranes used in commercial thin-film composites are more complex, containing interconnected networks of crumples[20] and clustered inner nanovoids, where the existing morphometry analysis cannot easily apply. Thus, understanding of the formation of this complex nanomorphology with spanning voids in 3D and how the nanomorphology determines membrane properties such as separation performance and mechanical properties have been unexplored.

Here, we present results elucidating the morphology–function relationships of PA membranes with clustered interconnected nanovoids by integrating experiments and simulations with graph theory (GT). By combining low-dose electron tomography with our customised morphometry analysis[13] adapted for interconnected PA membranes, and 3D void reconstruction, we quantitatively compare the nanomorphology of interconnected PA membranes synthesised with three different conditions. The local thickness and void size of the PA membranes−both determining the diffusion paths of solvent molecules across the membranes[5,19]−are influenced by the monomer concentrations. An intriguing observation of oligomer coalescence-and-growth during the nanovoid network formation process is supported by 3D local thickness mapping and coarse-grained molecular dynamics (CGMD) simulations of the interfacial polymerization (IP) process. Membrane performance is found to depend on a collective set of nanomorphological parameters, such as nanovoid types, surface area of the 3D voids, local membrane thickness and the molecular-level degree of crosslinking (DOC), all of which are shown to be tuneable by monomer concentrations. To comprehensively describe the networked nanovoid and crumple structures of the PA membranes, we couple skeletonization of the tomograms with GT, which accounts for arbitrary structural complexities, and together have been recently applied to describe materials networks from two-dimensional (2D) images[21,22]. We apply it to soft materials based on electron tomography. The extracted GT-based graph density and efficiency correlate with the apparent modulus of the membranes, confirmed by atomic force microscopy (AFM) mapping. Beyond membranes, the developed void reconstruction and GT-based representation methods can be applied to other complex nanomaterials, including biological structures such as protein assemblies[23], liposomes and synthetic materials such as polymer composite gels[24], porous hydrogel networks[25], block copolymers[26,27], nanoparticle assemblies[28] and energy storage materials[11].

## Results

### 3D reconstruction of PA membranes with interconnected nanovoids

PA membranes with interconnected nanovoids and crumples prepared using three synthetic conditions are studied to understand the 3D nanomorphology and relate the morphologies with their separation and mechanical properties. PA membranes are synthesised via IP of trimesoyl chloride (TMC) containing acyl chloride groups in an organic phase (hexane) and m-phenylenediamine (MPD) containing amine groups in an aqueous phase, on a polysulfone (PS) support and a cadmium hydroxide nanostrand (NS) sacrificial layer following prior work[29] with adjusted monomer concentrations (Fig. 1a). Due to the limited solubility of TMC monomer in the aqueous phase and the comparatively higher solubility of MPD in hexane, the MPD monomer diffuses into the organic phase reacting to form crosslinked PA membranes in the organic phase, but in proximity to the interface[30]. The resulting free-standing membranes are thin films with an apparent height below ~400 nm (Supplementary Fig. 1). For the three samples of our focus, the concentration of the MPD monomer ($c_{MPD}$) and the reaction time are maintained constant at 5 w/v% and 60 s, respectively, whereas the concentration of the TMC monomer ($c_{TMC}$) is varied from 0.05 (PA1), 0.1 (PA2) and 1 w/v% (PA3) (Fig. 1b, Supplementary Fig. 1, Supplementary Movie 1). Our previous work showed that high $c_{MPD}$ results in PA membranes with interconnected crumples, similar to those used in commercial applications[31]. Previous work on PA thin-film membranes used comparable monomer concentrations and reported methanol permeance values more than 20 times as those of commercial thin-film membranes[29]. Two membrane samples from each synthetic condition are examined to validate consistency for quantitative comparison (Supplementary Fig. 2). Furthermore as a comparison, control PA membranes are synthesised without the use of a PS support layer following the same starting monomer concentrations and reaction time, to demonstrate the consistency of the observed interconnected and nanovoid-containing morphology (Methods, Supplementary Note 1, Supplementary Fig. 3).

Low-dose electron tomography based on TEM (with a dose rate of 4−7 e⁻Å⁻²s⁻¹) is used to image the 3D nanomorphology of the synthesised PA membranes (Methods). In our electron tomography experiments, the sample is tilted along the x-axis from –60° to +60° in 2° intervals, to obtain 61 projections (Supplementary Fig. 4). By combining these tilted projections, we generate a tomogram, which is a 3D reconstruction of the PA membranes. The greyscale intensity-based reconstructions for the six membranes of the three synthetic conditions, PA1, PA2 and PA3, each with a voxel size of 3.5 Å, are shown in Fig. 1c. As evident from both 2D images and the 3D reconstructions, the PA membranes have distinct membrane morphologies, with the crumple density increasing with the increase in $c_{TMC}$ from 0.05 to 1 w/v% (i.e. PA1 to PA3). This observation is consistent with our previous work on PA membranes with spatially separated crumples, in which we showed that the density of crumples is dictated by the monomer concentrations in a manner consistent with the origin of crumpling being a reaction−diffusion instability. The reaction−diffusion instability predicts that at constant $c_{MPD}$, increasing $c_{TMC}$ results in denser crumples[13]. In addition to mapping the heterogeneity of morphology in the xy-plane, Fig. 1d depicts the 3D volume reconstructions at selected regions of PA1, PA2 and PA3 and the corresponding slices along the z-axis, showing that the crumples form networks throughout, spanning the 3D space with interconnected inner nanovoids. Comparing the tomographic volume reconstructions with conventional projected TEM images (Supplementary Fig. 5), the 3D structure of the membranes, especially the inner voids, cannot be justifiably captured by the latter.

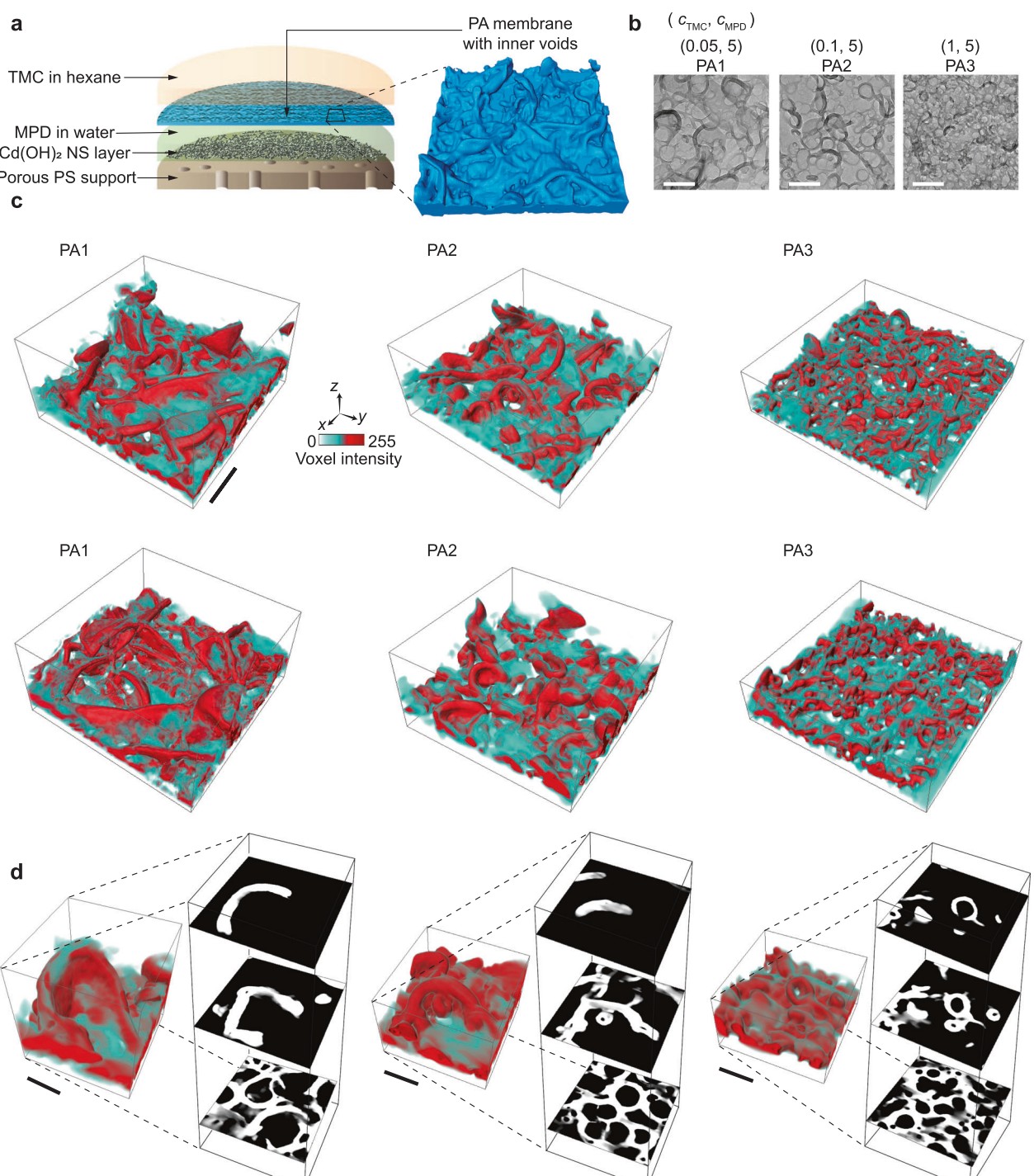

**Fig. 1 | Synthesis and tomographic reconstruction of PA membranes with interconnected crumples. a** IP of MPD in aqueous phase and TMC in organic phase on the sacrificial and porous support layers. The schematic is zoomed-in (black box and dotted lines) to show the interconnected crumple morphology. **b** TEM images for PA1, PA2 and PA3 synthesised with 0.05, 0.1 and 1 w/v% $c_{TMC}$, respectively, at a fixed $c_{MPD}$ of 5 w/v% and a reaction time of 60 s. **c** Greyscale intensity-based volume reconstructions for two membranes each of PA1, PA2 and PA3. **d** Greyscale volume rendering and corresponding $xy$-orthogonal sliced views across networked crumpled regions of PA1 (left), PA2 (middle) and PA3 (right) showing morphology change across the $z$-axis. The bottom $xy$-slices of the three membranes show an interconnected, networked morphology. Scale bars: **b**, **c** 200 nm and **d** 50 nm.

## Morphometry of local thickness to understand morphogenesis

The interconnected morphology and spanning 3D voids of PA membranes give rise to varying local thickness across the membrane and are associated with corresponding variations in the permeation length and local transport during filtration[19,32]. Geise et al.[33] demonstrated an example of where accurate membrane thickness values are needed by using solution–diffusion theory, which requires knowledge of the thickness of the active layer, to model the upper bound of the inverse correlation between water permeability and salt selectivity in PA thin-film desalination membranes[20,33]. Similar to our previous work mapping the local thickness of separated segmented crumples[13,19], our tomography-based method allows for direct mapping of the local thickness of PA membranes with networked crumples, excluding the 3D nanovoid spaces, thereby presenting a

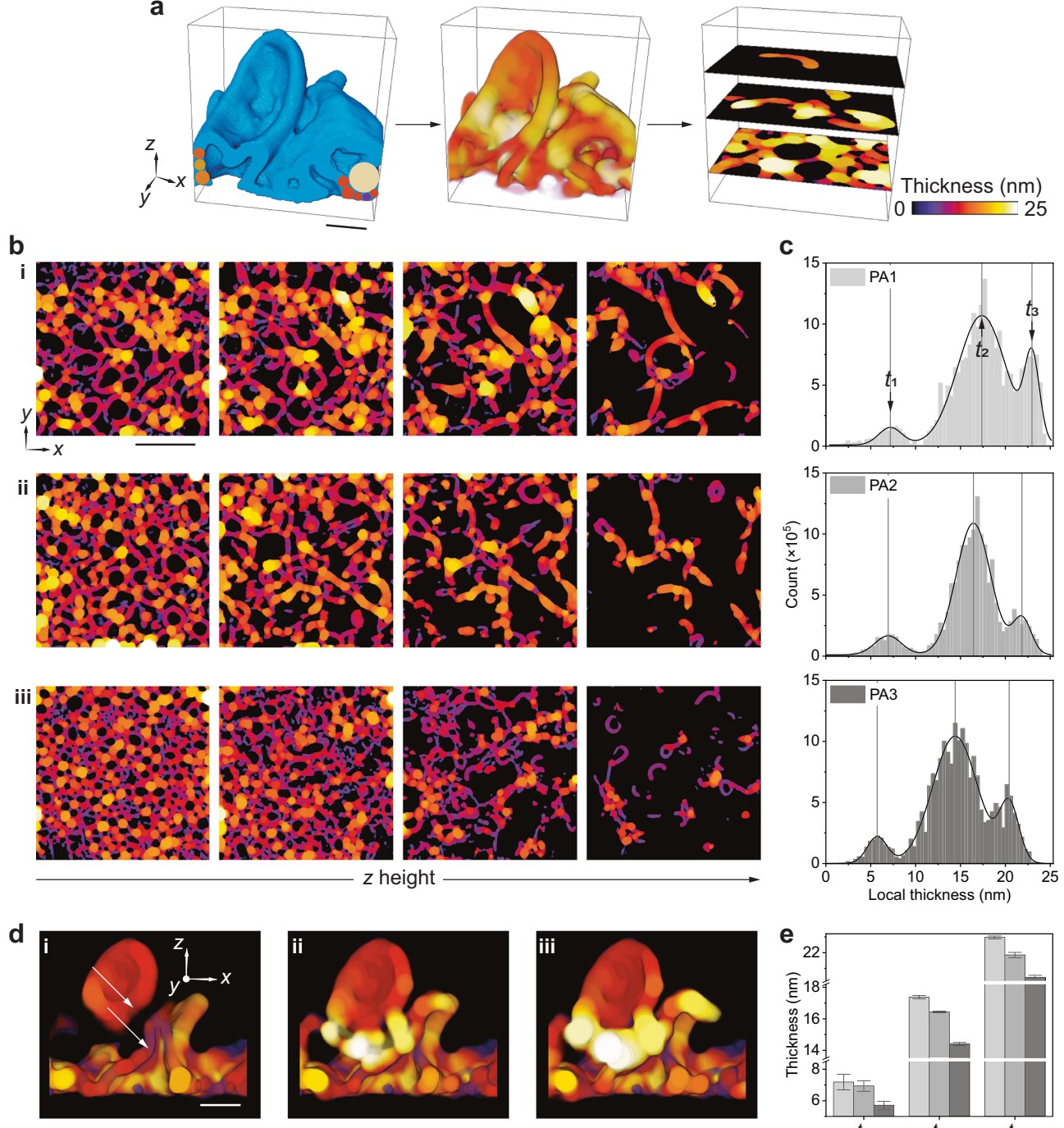

**Fig. 2 | Trends in local thickness reveal insights on the morphogenesis of interconnected nanovoids and crumples. a** Schematic showing volume reconstruction cross-section with inscribed spheres used to measure local thickness and sliced views of local thickness maps. The tomographically reconstructed 3D volume is filled with spheres (left) to generate a 3D thickness map (middle). The thickness map can be viewed as slices along the height (right) to observe local thickness changes and voids. **b** Representative *xy*-plane slices of the local thickness maps for (i) PA1, (ii) PA2 and (iii) PA3 coloured to the diameter of inscribed spheres. **c** Local thickness distributions for PA1 (top), PA2 (middle) and PA3 (bottom) with Gaussian fitting to show multimodal distributions with three thickness maxima, $t_1$, $t_2$ and $t_3$. The crumple wall with the lowest thickness ($t_1$) is the least abundant in all three PA membrane systems. **d** The *xz*-sliced cross-sections along two crumple walls showing two thin regions (white arrows in i) merging to form a thicker region where the two walls touch as the slices progress through (i) to (iii). The colour map legends for **b**, **d** are the same as **a**. **e** Trends in $t_1$, $t_2$ and $t_3$ local thickness values for the three membranes. The chart legend is the same as **c**. All three local thickness values decrease from PA1 to PA3. The error bars are from peak maxima fitting using Gaussian model (Supplementary Table 1). Scale bars: **a**, **d** 30 nm and **b** 200 nm.

method for accurate active layer thickness measurements for solvent permeability calculations.

Here, the local thickness of the membrane is defined as the diameter of the largest sphere inscribed within the membrane containing a given voxel[34] as shown in Fig. 2a. Using this definition, the local

thickness of the three types of PA membranes is mapped and representative *z*-slices of the tomographs are presented in Fig. 2b. As evident from the local thickness maps, the PA membranes show variations in local thickness across all axes (Supplementary Movie 2). The local thickness of the membrane is greater in regions where two or

more crumple walls meet as opposed to the regions where the walls do not touch (Fig. 2b).

We observe a multimodal distribution with three maxima ($t_1$, $t_2$ and $t_3$) in the local thickness histograms of all the membranes synthesised at different conditions (Fig. 2c), showing less abundant thin membrane walls ($t_1$) and more abundant thick regions ($t_2$ and $t_3$) (Supplementary Fig. 6, Supplementary Table 1). Previous studies of PA membrane cross-sections using scanning electron microscopy (SEM) measured average thickness values of 14–30 nm[20], which align with $t_2$ and $t_3$ local thickness values reported here. However, unlike previous work, the tomography-based method that we adopt is capable of mapping the local thickness of less abundant thin-walled membrane regions ($t_1$, Fig. 2c), showing local thickness values of 5–10 nm.

The observations of thicker walls where two or more crumples meet and the multimodal thickness maxima suggest that the characteristic interconnected crumples of these PA membranes can be potentially formed by local, physically contacting regions or by coalescence of growing crumples. Closer investigation of the $xz$-sliced volumes across the crumple walls shows that without the crumples touching, the walls are thinner ($t_1$) than the slices where the crumples touch ($t_2$ and $t_3$) (Fig. 2d, Supplementary Movie 3). Moreover, as shown in Fig. 2e, the second ($t_2$) and third ($t_3$) local thickness maxima are approximately twice and three times the value of the first ($t_1$), supporting that two or more crumple walls connect or merge to form thicker local regions ($t_2 \approx 2t_1$ and $t_3 \approx 3t_1$, where $t_1 = 7.2 \pm 0.5$, $6.9 \pm 0.3$ and $5.7 \pm 0.3$ nm, $t_2 = 17.4 \pm 0.1$, $16.4 \pm 0.1$, $14.4 \pm 0.1$ nm, $t_3 = 23.0 \pm 0.1$, $21.9 \pm 0.2$, $20.4 \pm 0.1$ nm respectively, for PA1, PA2 and PA3; Supplementary Table 2). Compared to the non-networked PA membranes in our previous work[13,19], the networked membrane has more peaks as the regions intercept multiple times now that the crumples are no longer spatially separated. Further assessing $t_1$, $t_2$ and $t_3$ values of the three PA membranes show that the local thickness decreases as $c_{TMC}$ increases (Fig. 2e), indicating that monomer concentration can be used as a means of controlling membrane thickness[13,31]. The observed decrease in local thickness with the increase of $c_{TMC}$ can be explained by decreases in the DOC of the PA membranes (Supplementary Fig. 7, Supplementary Table 3). Previous studies showed that the PA membranes with higher DOC are thicker than those with lower DOC[35].

## Coalescence and growth mechanism of nanovoid and crumple formation

Electron tomography-based local thickness mapping and CGMD simulations (*vide infra*) reveal a coalescence and growth mechanism governing the nanomorphogenesis of membrane crumple networks, alongside other morphogenetic processes. In previous studies, there are considerable discussions on the formation of PA membranes with either spatially separated or networked crumples[20,29,36,37]. One such hypothesis is that during IP, PA first forms a flat membrane and blocks the monomers from reacting further, acting as a self-limiting barrier[20]. The subsequent protrusion and crumple formation has debated origins ranging from interfacial boiling[36], different rates of monomer diffusion[20], or increases in local temperature[29]. In addition, Tang and co-workers propose that nanovoids and crumples form as a result of nanobubbling of carbon dioxide from the heat and hydrogen ions generated during IP[38,39]. A contrasting hypothesis claims that instead of a self-limiting barrier, an incipient flat membrane with pores is formed at the interface[37]. The crumple structure is attributed to the amine monomer diffusing through the pores and reacting with the acyl chloride monomer in the organic phase[37]. Eventually, the pores are filled to form a continuous membrane[37]. These hypotheses are directly linked to the high rate of product formation and IP reaction with characteristic dimensions at the nanoscale, which is challenging to probe experimentally[40]. Hence, the different observations and conclusions of morphology formation might be a result of different methods of synthesis and simulation conditions. Our previous work on PA

membranes of spatially separated crumples showed that a Turing instability-like diffusion-reaction accounts for many of the characteristics of PA membranes prepared by IP: (i) the protrusion of the IP reaction front ultimately into crumples and (ii) more quantitatively, their average lateral separation, or wavelength, which follows a power law dependence on the monomer concentrations[13]. We proposed that the final film morphology corresponds to the point at which the Turing instability-like phenomenon of the reaction front is frozen by the eventual coalescence of a contiguous membrane that blocks further diffusion of MPD into the hexane phase[13]. However, some aspects of the film morphology, such as variations in local thickness—a direct result of the 3D nanovoids embedded within the crumples—were not fully accounted for by the interfacial instability analysis.

To gain a better understanding of the formation mechanism, we integrate electron tomography with CGMD simulations, allowing us to characterise the mechanism of nanomorphogenesis from the early stages of IP reaction. It is worth noting that there are various simulation approaches available to describe the formation of the IP process of the PA membrane, mostly based on atomistic simulations (details in Supplementary Note 2). While PA membranes constructed at the atomistic-level offer a close representation and prediction of local characteristics such as dry and hydrated densities, and molecular transport properties, these models are computationally expensive, often used for computational cell sizes of ~5–20 nm in in the $x$ and $z$ dimensions[41,42]. As a result, these simulations cannot accurately reproduce experimentally observed membrane morphologies, which can have voids and structures on the scale of 100s of nm and require longer timescales to form. In our work, we use the CGMD model developed by Muscatello et al.[43] to overcome the limitations. This CGMD approach utilises a multi-scale modelling technique, where the CG models maintain the shape and connectivity of the monomers, while simulating monomer diffusion without explicitly modelling the solvents themselves. Additionally, it offers the ability to map these CG models onto fully atomic configurations through a subsequent relaxation procedure. By employing this multi-scale modelling technique, we can achieve accurate representation of the monomer dynamics and bond formation while still benefiting from the computational efficiency provided by CGMD modelling.

For the CGMD simulations, the monomers are coarse-grained into representations, where the bond lengths, molecular conformations as well as reactivity are consistent with those of a real atomistic structure. The reacting MPD and TMC monomers are represented as a triad of beads arranged in a rigid equilateral triangle with pendant reactive groups (Fig. 3a). These are referred to as CG-MPD and CG-TMC monomers, respectively. The reacted unit, known as the TMC/MPD dimer, reacts to form CG-TMC/MPC oligomers, which bridge CG-MPD and CG-TMC monomers via an explicit amide bond. Next, we set up the computational cell such that the initial distribution of CG-TMC and CG-MPD is phase-separated (gridded) following experimental conditions, and IP starts as we switch off the repulsion between the grids and CG-MPD monomers, allowing them to diffuse, mix with CG-TMC, and form into PA membranes (Fig. 3b). By varying ratios of CG-MPD and CG-TMC monomers in the reservoirs, we can achieve the desired stoichiometry during the IP process. Thus, this approach facilitates simulations that capture the essential stereochemical features of the monomers and the amide bonding between them. We employ two different TMC:MPD ratios in our simulations. One is the experimental PA3 membrane with a 1:12.3 mole ratio (1 w/v% $c_{TMC}$ and 5 w/v% $c_{MPD}$). Additionally, a PA membrane (PA4) with a computationally less expensive model of 1:4 (2 w/v% $c_{TMC}$ and 3.26 w/v% $c_{MPD}$) mole ratio is used to supplement the findings from PA3 (Fig. 3c, Supplementary Figs. 8,9). These ratios are chosen to mimic the experimental IP procedures and explore the formation of PA membranes and their interconnected crumpled morphologies.

Our CGMD simulations show that PA membranes formed with two different TMC:MPD ratios exhibit similar growth mechanisms and in

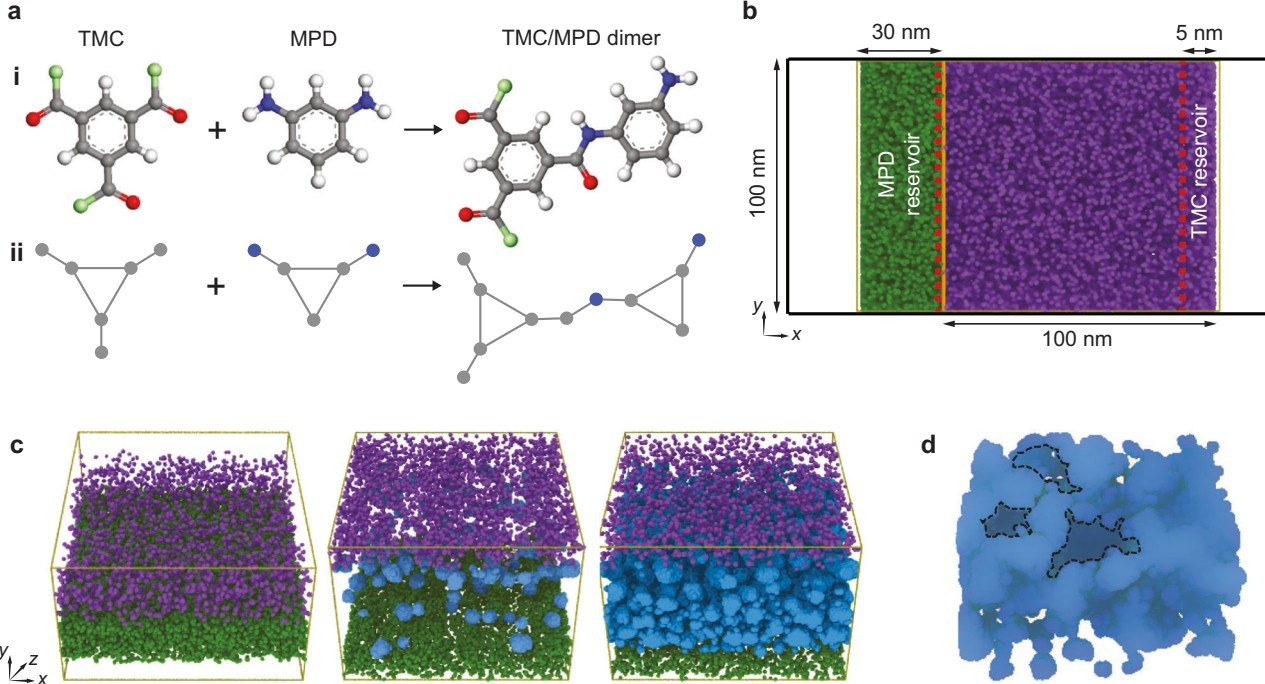

**Fig. 3 | CGMD simulations on the mechanism of nanovoid and crumple formation. a** Depictions of (i) atomistic and (ii) CG structures for TMC monomer, MPD monomer, and reacted TMC/MPD dimer. Note that carbon, nitrogen, oxygen, and hydrogen atoms are shown as grey, blue, light green and white circles, respectively. **b** Simulation setup showing the initial distribution of TMC (purple) and MPD (green) monomers within the simulation cell, with the reservoir boundaries depicted in red dashed lines. To prevent the TMC and MPD monomers from coming into proximity, a repulsive potential represented by a yellow line is employed. To prevent their interaction due to periodic boundary conditions, two 25.0 nm vacuum layers are incorporated on two sides. **c** Based on a TMC:MPD monomer ratio of 1:12.3, the snapshots showcase the formation of a spanning membrane in a CGMD simulation at three successive times: 0 ns, 90 ns, and 600 ns. These snapshots are arranged from left to right, providing a visual depiction of the progressive stages of membrane formation over time. Best perspective views are selected to show the oligomer and spanning membrane distributions. The simulation box (100 × 130 × 100) nm³ at 0 ns shows the initial distribution of TMC (purple) and MPD (green) monomers before permitting IP reactions. **d** Zoomed-in region of the spanning membrane showing nanovoids (shown as grey regions within black dashed lines). Scale bar: **d** 4 nm.

agreement with previous work[43] show that the membrane formation process can be described by a combination of self-limiting reaction–aggregation process and reactant diffusion. In the CGMD simulation, we observe that the PA membrane shapes through the formation and coalescence of oligomer clusters at the interface with three key structural features. Firstly, while the simulation maintains a constant MPD:TMC ratio, a single CG-MPD monomer yields two amine groups, whereas one CG-TMC monomer contributes three carboxyl groups. Consequently, there is an imbalance in the number of reactive groups of CG-MPD and CG-TMC. Secondly, the PA membrane consists of regions where the degree of polymerization is high, interspersed with interfaces where the degree of polymerization is relatively low. Lastly, significant variations in local density are observed, including voids. These structural features captured in the CGMD simulations align with our experimental results and previous work using ensemble spectroscopy methods[44]. Specifically, taking the 1:12.3 TMC:MPD ratio (PA3) as an example, during the early stages of PA membrane formation, CG-MPD monomers diffuse into the region where CG-TMC monomers are restricted. Some of these monomers react to produce small CG-TMC/MPD oligomers, which then undergo further polymerization and aggregation (Fig. 3c, Supplementary Fig. 8). Over time, a clustered network is formed by coalescence of different oligomers, which continues to grow as additional monomers and/or oligomers accrete at reaction sites on the membrane surfaces (Fig. 3c) until the reactants are depleted or the membrane becomes continuous that self-limits the growth.

The CGMD simulation also allows us to track the spatial variations of surface chemistry during the IP process. Interestingly, we find that prior to the coalescence of oligomer clusters, unreacted acyl chloride

and amine side groups tend to reside predominantly on the surfaces of the clusters. Once two clusters coalescence, it becomes challenging for the unreacted side groups on the cluster surfaces to undergo further reactions due to steric hindrance or an imbalance in the number of amine and carboxyl reactive groups. As a result, the interfaces between oligomer clusters can have sites that are not reacted or forming into amide bonds, leading to the formation of voids in the PA membranes (Fig. 3d, Supplementary Fig. 10). Note that in our CGMD simulations, the system is maintained at a relatively constant temperature under Langevin dynamics, which is limited in modelling local effects such as reaction heat and interfacial boiling. Nevertheless, the match between the PA membrane morphologies from the CGMD simulation and our experiments highlights that the monomer diffusion-reaction characteristics and the resulting oligomer coalescence-and-growth mechanism determine the key morphological features of the PA membrane. We also incorporate additional simulations to investigate the impact of amine monomer diffusion rates on the IP process, where denser membrane structures are formed when the diffusion rate of the amine monomer is low (Methods, Supplementary Note 3, Supplementary Figs. 11, 12).

Overall, our CGMD simulations along with our experimental observations of multimodal local thickness increments support the mechanism of oligomer coalescence and the growth process of morphogenesis of interconnected PA membranes, supporting that the key morphological features of the PA membrane (i.e. voids and local thickness) are determined by reaction–diffusion instabilities. Our tomographic capability to detect the full local thickness distribution provides evidence for oligomer coalescence-and growth in real experimental membranes—by showing that thicker interconnections

are formed by merging of regions of thinner thicknesses—supporting previous simulations on membrane formation[43].

## Volume-filled nanovoid reconstruction to theoretically fit permeance

Membrane thickness is an essential input parameter for theoretical predictions of membrane performance[33,45]. Beyond membrane thickness, the most notable feature of the networked PA membranes is the presence of extensive clustered voids spanning the full 3D volume (Fig. 4a), which are clearly revealed by electron tomography. The presence of inner voids in filtration membranes has been shown to influence the water permeability of PA membranes[5,20,37,46]. Prior work has focused on characterising the voids by visualising the cross-sectional views of the membranes using TEM and SEM[14,47–49] or by estimating the void volume by water uptake measurements or spectroscopic characterisation of effective refractive indices[45,50]. Previous work by Pacheco et al.[14] presented two types of voids in PA membranes; however insufficient contrast between the PA material and background in their tomograms led to only a partial, manual reconstruction of the voids[14]. Culp et al.[15]. demonstrated that closed voids can be observed in tomogram cross-sections[15]. Our group has previously delineated the inner voids enclosed by individual crumples in a PA membrane as low TEM contrast regions through material reconstructions[19]. However, these efforts do not render the complete 3D shape or clustering of the voids.

We develop a workflow of void analysis method to map the clustered nanovoid spaces underlying the interconnected crumples. As shown in Fig. 4a, b, our interconnected PA membrane has complex 3D inner voids, which are challenging to resolve using the material reconstruction alone. To overcome the challenge of segmenting voids from background, we use the PA material as a boundary between the background and the voids ("Methods"). In this way, the void spaces are partitioned and binarised with the background removed, enabling volume-filled reconstruction of the voids (Fig. 4b, c). This workflow allows us to demonstrate a complete 3D nanovoid map reconstruction for PA membranes (Fig. 4d, Supplementary Fig. 13, Supplementary Movie 2), which enables direct extraction of the quantitative features of the voids in a PA membrane, such as void volume, void ratio ($r$) (i.e. the percentage fraction of void volume to material volume) and void volume fraction (the percentage fraction of void volume to total volume of voids and material). Specifically, we discern two types of voids—open and closed as defined in prior work[14]—in our interconnected PA membranes, where open voids have a pore opening to the undersurface of the PA membrane and closed voids are completely enclosed by the PA material (Fig. 4c). The presence of such open and closed voids in PA membranes creates distinct tortuous transport pathways for solvent, which is critical for modelling permeance[37,46].

The concentration of TMC monomer is shown to serve as a handle to tune the morphology (e.g. volume, spatial distribution) of voids in the PA membranes. We use our volume-filled void reconstruction method to map the void distribution of the three PA membranes in a projection area of 727 nm × 727 nm to reveal archipelago-like nanovoid maps (Fig. 4d, e). Our results show that at a constant $c_{MPD}$, when $c_{TMC}$ increases from 0.05 to 1 w/v% (from PA1 to PA3), the volume of PA material ($v_{tm}$) decreases from $(49.2 \pm 4.0) \times 10^{-3} \, \mu m^3$, $(42.4 \pm 5.7) \times 10^{-3} \, \mu m^3$ to $(32.4 \pm 3.4) \times 10^{-3} \, \mu m^3$ and the volume of the void spaces ($v_{tv}$) decreases from $(12.7 \pm 1.2) \times 10^{-3} \, \mu m^3$, $(8.7 \pm 0.9) \times 10^{-3} \, \mu m^3$ to $(6.1 \pm 0.3) \times 10^{-3} \, \mu m^3$, respectively. A similar trend is observed in $r$, ($r = v_{tv}/v_{tm}$), where $r = 25.7\% \pm 0.4\%$, $20.5\% \pm 0.6\%$ and $19.0\% \pm 1.1\%$ for PA1, PA2 and PA3, respectively (Fig. 4f, Supplementary Table 4). The void spaces of the three membranes contribute to $20.5\% \pm 0.2\%$, $17.0\% \pm 0.4\%$ and $16.0\% \pm 0.8\%$ of the total volume in PA1, PA2 and PA3 respectively (Supplementary Table 4), within the range of the void fractions (15% to 35%) previously reported for PA membranes using spectroscopic methods and microscopic

characterisation[45,49,50]. Furthermore, our volume-filled void reconstruction method allows us to separately visualise the individual voids in the projected 3D nanovoid maps (Fig. 4e). As seen in Fig. 4d, e, PA1 has large, clustered nanovoids spanning through the reconstructed volume compared to PA2 and PA3. PA1 has the lowest $c_{TMC}$, hence the largest separation between crumples. As discussed in Fig. 1c, the density of crumples is governed by a reaction–diffusion instability, resulting in higher crumple density at high $c_{TMC}$, and vice versa. PA1 has the lowest $c_{TMC}$, hence the largest separation between crumples, resulting in the lowest crumple density of the three membranes. Therefore, during membrane formation, the reaction front in PA1 deforms into a larger region compared to the crumples of PA2 and PA3, leading to larger voids. PA3 on the other hand, has the highest $c_{TMC}$ and the highest crumple density among the three membranes, resulting in smaller crumples and smaller voids. Furthermore, due to the role of voids in solvent permeance[5], there have been extensive efforts to increase the void volume within PA membranes, from using porous materials such as zeolitic imidazole frameworks[5] to employing void-tailoring agents[37] and even nano-foaming with the aid of ultrasonication[51]. As previous work has demonstrated[52], we show that monomer concentration alone is also a good strategy to increase void volume.

This level of detail in PA membrane void reconstruction paves the way for better understanding the effect of nanoscale void morphology on solvent permeance. PA membranes are employed in pharmaceutical, chemical and petrochemical industries to separate and recover organic solvents used as raw materials or cleaning agents in a process called organic solvent nanofiltration, which can separate organic impurities in the range of 200–1000 Da from organic solvents[29,53]. The use of polymer films, such as PA membranes for organic solvent nanofiltration is promising because it is an energy-efficient and cost-effective process[17]. Therefore, to better understand the separation behaviour of organic solvents, we explore the permeance of methanol across the three PA membranes under study.

Our previous work used the Spiegler–Kedem model to predict the methanol permeance of simpler PA membranes with separated crumples[13]. As to inputs for the model, our previous work used the DOC of the membranes, surface areas of crumples and flat membrane regions and assumed that the membrane thickness is uniform[13]. Here this work clearly shows that the membranes as a whole consists of interconnected voids without flat membrane regions and that the local thickness of the PA membranes exhibits multimodal distributions due to formation of nanovoids; as a result, we modify the inputs of the Spiegler–Kedem model: we still include the DOC of the membranes, but consider the variations in local thickness over the membranes, the areas of the top and bottom surfaces of the membranes and additionally the 3D surface areas of open and closed voids (Supplementary Table 5) to comprehensively capture the effect of the nanomorphology (Approach 1). Specifically, the 3D surface area of voids is calculated by enveloping open and closed voids with a triangular mesh surface. In our model, we assume that the open voids increase the surface area for solvent transport, whereas closed voids created additional barriers for solvent transport. The DOC of the membranes determined by X-ray photoelectron spectroscopy (XPS) (Supplementary Fig. 7, Supplementary Table 3), is 97.7 %, 73.9 % and 41.8 % for PA1, PA2 and PA3, respectively, similar to values reported in previous literature ranging from 42 to 99%[29,54]. With these inputs for the Spiegler–Kedem model[55], we develop theoretical fittings for the methanol permeance across the membranes (Methods, Supplementary Note 4). A fitting which uses the apparent membrane thickness measured without considering inner voids by AFM (Approach 2)—a commonly used characterisation method to measure PA membrane thickness[29]—is presented for comparison (Supplementary Figs. 14, 15). Approach 1 provides qualitatively better theoretical fittings for the experimentally measured methanol permeance compared Approach 2 (Fig. 4g) both in terms of the trend and the permeance values. It should be noted that the experimentally

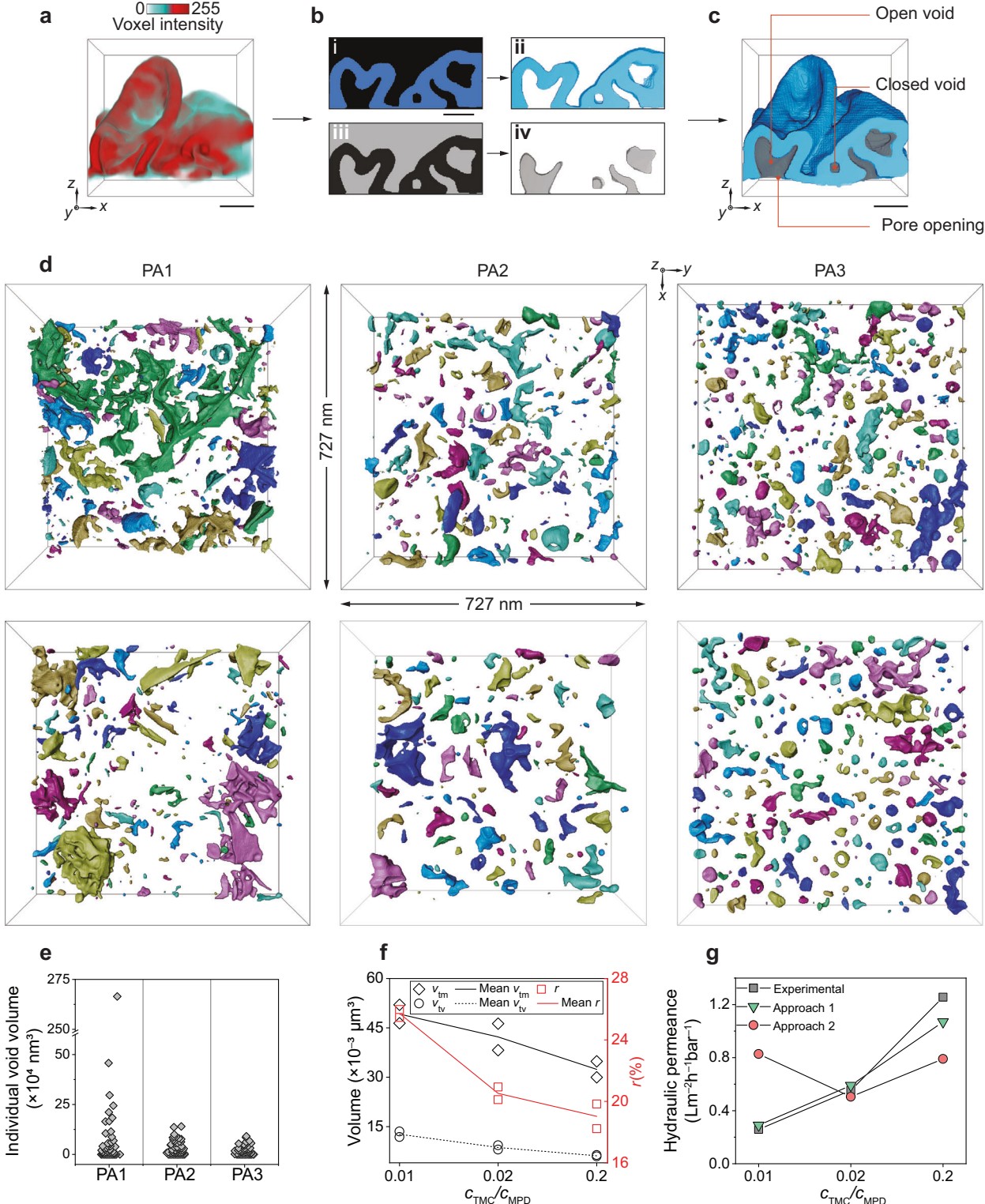

**Fig. 4 | Visualising and quantifying 3D nanovoids and their relationship with separation performance. a** Greyscale-based reconstruction showing PA membrane region with crumpled morphology. **b** Workflows of PA material (blue) and void (grey) extraction using *xz*-slices of the tomogram. In PA material rendering, the blue region of the tomogram is binarised (i) and the PA material (ii) is reconstructed. In the void reconstruction, binarisation (grey region) includes both voids and background (iii). Hence, the continuous PA membrane is used as a boundary to delete the background and reconstruct the voids (iv). The volume rendering images (ii, iv) show a four-slice thick volume for PA material and void, whereas binarised images (i, iii) are single slices. **c** Reconstruction showing two types of voids. Open voids have a pore which opens the void at the undersurface of the membrane and closed voids are surrounded by PA material. **d** Archipelago-like void maps of PA1, PA2 and PA3 depicting the void distribution of the PA membranes in a projected area of 727 nm × 727 nm. Void maps are assigned colours randomly to clearly show separate individual void islands. **e** Individual void volume distributions for PA1, PA2 and PA3. **f** PA material volume ($v_{tm}$) and void volume ($v_{tv}$) of the PA membranes (left *y*-axis, black), and fraction of void and material volumes ($r$, right *y*-axis, red). The $c_{TMC}/c_{MPD}$ ratios of 0.01, 0.02 and 0.2 represent PA1, PA2 and PA3 respectively. The mean values of $v_{tm}$, $v_{tv}$, and $r$ for multiple PA membranes synthesised at the same condition are shown by line plots ($n = 2$). **g** Experimental measurements and theoretical approaches for methanol permeance. Scale bars: **a–c** 30 nm.

measured methanol permeance of the PA membranes in this study is within the same order of magnitude of similar membranes studied in previous work[13,29]. Using the theoretical fittings, our results show that the nanomorphological parameters−local thickness and nanovoid surface area−are important in providing estimations of organic solvent permeance across the PA membrane, compared to AFM based measurements.

## GT representation relationship to apparent nanomechanical moduli

Although the void analysis and local thickness mapping facilitate quantitative comparison among PA1, PA2 and PA3 on the morphology formation mechanism and separation performance, these measures are local geometric properties and do not capture the overall structural rigidity of the membrane. Thus, we adopt GT analysis to the electron tomographic reconstruction. Recent work has started to use GT to describe materials systems such as organic molecules[56], gold nanoplatelet-based colloids[57], tetrahedral gold nanoparticle pinwheel superlattices[21], colloidal networks[58], enzymes and virus particles[59] as a set of nodes and branches, although a major challenge is to obtain a graph that represents the nanoscale structural relationships of the starting material[22,60]. Our GT analysis shows that the structural rigidity trends monotonically to the mechanical robustness of the membranes, which is an important property because the PA membranes used in filtration and reverse osmosis need to endure high hydraulic pressures[29].

We use skeletonization to construct a structural graph using the reconstructed 3D volumes of PA membranes. In previous studies, skeletonisation has been used to simplify complex 3D structures ranging from blood vessels in cancer tissue[61] to nanoporous gold[62] by minimising the dimension of an object while preserving its topology and geometric features[63]. A skeleton is composed of nodes, which are connection-points, branch-points, or end-points of a skeleton[64] and branches, defined as curved lines which connect nodes (Fig. 5a). In our case, the 3D volume from tomographic reconstruction is reduced to a set of one-dimensional curves in 3D space with a thickness of a single voxel (Fig. 5b). The resulting skeleton captures the morphology of the membranes−such as nanovoid size, nanovoid clustering, crumple interconnection, crumple density and membrane curvature (Fig. 5c)− by quantifying the 3D volume using the connections between nodes and branches. Figure 5b–d shows a zoomed-in section of PA2 skeleton graph and the skeletons of the three PA membranes (PA1–PA3) (Supplementary Movie 2). Both the numbers of the nodes and branches of the skeleton decrease as $c_{TMC}$ increases from PA1 to PA3 (Fig. 5e, Supplementary Fig. 16, Supplementary Table 6) given a fixed projection area (727 nm × 727 nm), likely a result of decreasing overall membrane volumes (Fig. 4f).

Further understanding of the networked skeleton requires more than statistics of nodes and branches, but geometric descriptions of the whole network, for which we employ GT. In previous work, different approaches were adopted in converting morphology into a quantitative graph; for example, microscale assemblies of nanoparticles were transformed into graphs by defining the nanoparticles and interparticle bonds as graph components[57] and grain boundaries and intergranular capacitance were used to discretise ceramic microelectronics into graphs[65]. In applying GT to the PA membranes, we use the skeletons as the input to describe the skeleton nodes and branches as a graph. Among the many GT-based parameters (Table 1, Supplementary Fig. 17, Supplementary Table 7), we find that graph density ($\rho$) and global efficiency ($E$) are the two metrics that we can use to compare structural similarity of graphical networks[22,66,67]. The GT metric $\rho$ describes the short-range organisation of the network, whereas $E$ deals with how efficiently the geometric network can transfer a given load[68].

These two GT parameters provide information in three aspects: (i) to quantify the PA skeleton graphs to show the similarity of PA membranes synthesised at the same condition and the difference among those at different conditions, (ii) to assess the interconnection and load transfer efficiency of the PA skeletons, thereby enumerating the graph rigidity and (iii) to find correlations between the apparent modulus of the PA membrane using the quantified structural connection and load transfer characteristics of the skeleton graphs. Figure 5f, g show that $\rho$ and $E$ are similar for skeleton graphs of the same condition and dissimilar across the three synthetic conditions, where $\rho = (9.0 \pm 0.2) \times 10^{-4}$, $(1.2 \pm 0.1) \times 10^{-3}$ and $(1.9 \pm 0.3) \times 10^{-3}$ and $E = (4.8 \pm 0.4) \times 10^{-2}$, $(6.1 \pm 0.4) \times 10^{-2}$ and $(6.9 \pm 0.7) \times 10^{-2}$ for PA1, PA2 and PA3, respectively. From these GT parameters and other morphology descriptors presented in this work, we conclude that the PA membranes with networked crumples of the same synthetic conditions are morphologically similar and their nanomorphology changes across the three synthetic conditions.

The GT parameters can be related to the mechanical properties of materials[22]. AFM is used to measure the apparent modulus of the three membrane systems. Here, the apparent modulus depends on the generic PA material modulus and nanomorphology. To map the local apparent moduli, the AFM tip is assigned to a matrix of evenly spaced locations over the surface to collect 256 × 256 force-indentation curves for each sample. The apparent moduli are calculated using Hertz model with corrections for substrate effects (Methods). As shown in Fig. 5h, there is heterogeneity in the modulus map for all three PA samples, arising from the irregular networked crumple nanomorphology. However, the mean apparent moduli show an increasing trend as $c_{TMC}$ increases from 0.05 to 1 w/v% (from PA1 to PA3) with values of $1.12 \pm 1.38$ GPa, $1.51 \pm 1.61$ GPa, $2.50 \pm 1.73$ GPa, respectively (Fig. 5i). These values are on the same order of magnitude as bulk measurements for PA membranes (ranging from 0.1−3.6 GPa) and our recent published work[13,29,69]. In GT analysis, a graph with high load transfer (i.e. high $\rho$ and high $E$) corresponds to a rigid skeleton and vice versa. Accordingly, PA1 is geometrically the least rigid skeleton with low interconnection and low load transfer compared to PA3, which has the highest $\rho$ and $E$, explaining the trend of apparent moduli. Thus, using GT parameters (i.e. $\rho$ and $E$), we show that skeleton rigidity relates to the trend in apparent moduli of PA membranes.

Furthermore, considering the nanomorphology descriptors presented in this work, void parameters show a large difference among the three membranes (Fig. 4f, Supplementary Tables 4,5), whereas the local thickness distributions are similar, regardless of the synthesis condition (Fig. 2c). Previous work theoretically calculated the Young's modulus for low dielectric constant porous materials, to demonstrate that as porosity decreases the Young's modulus increases[70]. To investigate the potential effect of void parameters in the modulus of the three PA membranes, we define the material fraction ($f_{material}$) and void fraction ($f_{void}$) as shown in Eqs. (1), (2).

$$f_{material} = \frac{v_{material}}{(v_{material} + v_{void})} \% \tag{1}$$

$$f_{void} = \frac{v_{void}}{(v_{material} + v_{void})} \% \tag{2}$$

PA1 has the highest $f_{void}$ and lowest $f_{material}$, with $f_{material}$ increasing and $f_{void}$ decreasing from PA1 to PA3, where $f_{material}$ and $f_{void}$ values are respectively, PA1 = 79.5 ± 0.2% and 20.5 ± 0.2%, PA2 = 83.0 ± 0.4% and 17.0 ± 0.4%, and PA3 = 84.0 ± 0.8% and 16.0 ± 0.8% (Supplementary Table 4). Considering the effect of voids alone on the modulus, PA1 with the highest void volume, should be easier to indent with the AFM cantilever. Thus, based on void parameters alone, the modulus is expected to adopt the trend of PA1 < PA2 < PA3, corroborating the GT-based analysis of skeleton rigidity with the mechanical properties of PA membranes. Note that PA membranes are complex materials with heterogeneity potentially even in DOC for the same sample, and the model discussed here illustrates the qualitative trend.

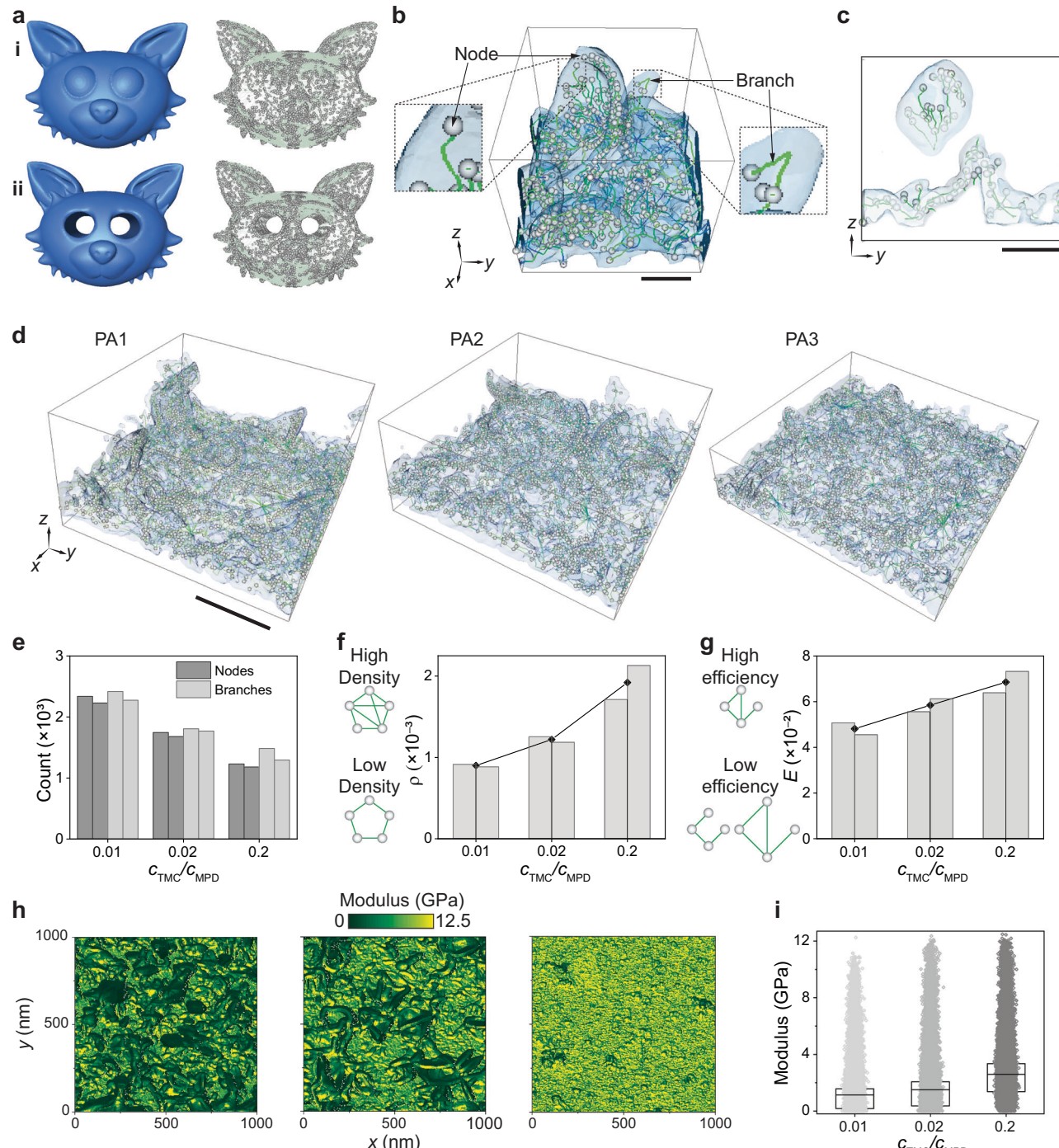

**Fig. 5 | GT applied to PA skeleton networks explains mechanical property trends. a** Cartoon depiction showing skeletonisation capturing the morphology of both (i) void-less and (ii) void-containing structures as well as their distinction. **b** Zoomed-in region of PA2 showing the skeleton resting within the translucent volume of the membrane material. The schematic shows the components of a skeleton. Nodes are depicted by circles and branches by lines. **c** Cross-section from schematic **b** showing that the skeleton captures the morphology of the membrane. **d** Skeletonization of PA1 (left), PA2 (middle) and PA3 (right). The PA material is shown in translucent blue encompassing the skeletons. **e** Plot depicting the number of nodes and branches for two PA membranes each from PA1, PA2 and PA3 ($c_{TMC}$/

$c_{MPD}$ ratios of 0.01, 0.02 and 0.2, respectively). GT metrics such as graph density $\rho$ (**f**) and global efficiency $E$ (**g**) show similarities between skeleton graphs of the same synthesis condition and contrast between the three conditions under study. The schematics on the left of the plots illustrate examples of simpler graphs with high and low metrics. The mean values of measurements for multiple PA membranes synthesised at the same condition are shown by line plots ($n = 2$). **h** Apparent modulus maps for PA1 (left), PA2 (middle) and PA3 (right). **i** Apparent modulus distribution for PA1, PA2 and PA3. The box range and mid-line are the interquartile range and mean values, respectively. Scale bars: **b**, **c** 30 nm and **d** 200 nm.

As a comparison with the mean apparent moduli measured from AFM, we develop three atomistic IP models for the PA membranes (Supplementary Fig. 18), using IP simulations described in previous work[71]. We subject the three models to compression loading

(Methods), monitor their corresponding stress-strain relationship and then extract their modulus (Supplementary Fig. 18). Notably, the calculated modulus values are 1.21 GPa, 1.48 GPa and 2.61 GPa, respectively, which closely match the experimental mean values. Likewise,

**Table 1 | GT parameters calculated for PA membranes with networked crumples**

| Parameter | Formula | Definition |
|---|---|---|
| Graph density | $\rho = \frac{2e}{[n(n-1)]}$ | The ratio between the number of branches in the graph to the total number of possible branches if all nodes were connected by a single branch[22,83]. |
| Global efficiency | $E = \frac{1}{n(n-1)} \sum_{i \neq j} \frac{1}{L_{ij}}$ | The efficiency of a pair of connected nodes is the reciprocal of the shortest distance between the nodes. The global efficiency of a graph is the average of efficiencies across all pairs of connected nodes[22,84] |

n: number of nodes.

e: the number of branches.

L: shortest distance connecting two vertices when travelling through edges.

the modulus of the PA membrane is improved as the TMC:MPD ratio increases. The molecular-level reason for this phenomenon can be that even though a higher TMC:MPD ratio results in a lower DOC as indicated in Supplementary Fig. 7, it facilitates the development of a more extensively crosslinked network structure due to the increased concentration of TMC monomers. The unreacted hydrolysed TMC monomers can form hydrogen bonds with carbonyl groups from either unreacted hydrolysed TMC itself or the PA membrane, further contributing to the strengthening of the network structure.

## Discussion

In summary, we use a 3D imaging and morphometry platform in combination with CGMD simulations and GT to elucidate the nano-morphology of highly complex PA membranes with interconnected nanovoids embedded within networked crumples. Using this multi-faceted approach, we present the relationship of nanovoids with membrane morphogenesis, methanol separation performance and nanomechanical properties in the form of the apparent modulus for PA membranes with nanostructures similar to those used in industrial applications. We combine 3D local thickness mapping and CGMD simulations of IP to uncover an oligomer coalescence-and-growth mechanism of interconnected void formation, inviting future discoveries on nanomorphogenesis of soft materials with irregular nanomorphologies, such as conductive polymers from interfacial oxidative polymerization[72], polymer membranes[73] and metal organic framework layers synthesised using IP[74]. Our volume-filled void analysis method not only allows us to visualize and quantify nanovoids within the PA membrane, it also solves the obstacle of practically reconstructing open pore structures in a nanoporous material, opening opportunities to uncover structure–property relationships in sponge-like and bicontinuous porous materials, such as collagen sponge scaffold-cell assemblies used in tissue engineering[75] and polymer-metal nanoparticle bicontinuous films used for stretchable electronics applications[76]. Moreover, understanding 3D arrays and networks of nanofibrils, biological filamentous systems and hydrogel networks, which are currently commonly quantified using surface characterisation or 2D projections[68], can be improved by applying our method of quantifying 3D volumes using skeletonization and GT parameters. For example, for the systems of our PA membranes, while the fundamental mechanism requires more future studies, the apparent modulus trend is shown to be related to the rigidity of skeleton network, which captures a myriad of morphology measures such as nanovoid size, nanovoid clustering, crumple interconnection, crumple density and membrane curvature. Broadly, we foresee that our work will provide additional avenues for understanding intricate irregular nanostructures and linking their functionalities with the morphology, granting future potential of harnessing this ability of morphology to guide properties and molding nanomaterials based on desired functions.

## Methods

### Synthesis of PA membranes

PA membranes were synthesised using a method described by Karan *et al*[29]. and used in our previous work[13,19,31]. Cadmium hydroxide

(Cd(OH)$_2$) NS were synthesised by stirring cadmium chloride hydrate (50 ml, 4 mM, 99.998%, CdCl$_2$·xH$_2$O, $x \approx 2.5$, Alfa Aesar) with ethanolamine (50 mL, 2 mM, >98%, Sigma-Aldrich) at 500 rpm for 15 mins, at which point the reaction mixture turned cloudy grey. A PS film (PS35, Sepro Corporation) soaked in Milli-Q water for 12 h prior to use, was used as a porous support in a glass filtration setup. Methanol (20 mL, 99.9%, Fisher Chemical) and Milli-Q water (50 mL, 18.2 MΩ·cm at 25 °C) were filtered through the PS support against a vacuum of −65 to −67 kPa. The Cd(OH)$_2$ NS solution was poured into the filtration setup and filtered under vacuum (−65 to −67 kPa) to form a sacrificial layer on the PS support film. An aqueous solution of amine-containing *m*-phenylenediamine (MPD, 10 mL, 5 w/v%, 99%, Sigma-Aldrich) was added carefully as to not disturb the sacrificial and support layers and was filtered under the same vacuum conditions. A solution of trimesoyl chloride (TMC, 98%, Sigma-Aldrich) in hexane (99.9%, Fisher Chemical), with concentration depending on the desired PA membrane morphology (0.05 w/v% for PA1, 0.1 w/v% for PA2 and 1 w/v% for PA3) was then poured into the filtration setup and was left to polymerize under atmospheric pressure. After 1 min, the reaction was stopped by pipetting out the TMC in hexane solution and washing three times with hexane (10 mL×3). With the support of the PS film, the synthesised PA membrane along with the Cd(OH)$_2$ NS layer was carefully transferred to float on the air-water interface of a Milli-Q water-filled petri dish, resulting in a free-standing membrane. The Milli-Q water was replaced with hydrochloric acid (HCl, 10 mM, 20 mL, 36.5−38.0%, Macron) and left overnight to dissolve the Cd(OH)$_2$ NS sacrificial layer. The dilute HCl solution was once again replaced with Milli-Q water, leaving the free-standing PA membrane floating in the air-water interface for further analysis.

The synthesis of control PA membranes (i.e. PA membranes without supporting PS layer) was performed using a method described by Cui et al.[77]. Control PA membranes were synthesised by carefully adding TMC in hexane, with concentration depending on the desired control PA membrane starting concentration (10 mL, 0.05 w/v% for control PA1, 0.1 w/v% for control PA2 and 1 w/v% for control PA3) along the wall of a glass petri dish containing an aqueous solution of MPD (10 mL, 5 w/v%). The control PA membrane was formed at the hexane–water interface. After 1 min, the reaction was stopped by carefully pipetting out the TMC in hexane solution (without damaging the fragile PA membrane) and washing three times with hexane (10 mL×3). The excess hexane was removed, and the membrane was transferred to float on the air–water interface of a petri dish with water.

### TEM imaging and electron tomography

Free-standing PA membranes were transferred onto a carbon-coated copper TEM grid (Electron Microscopy Sciences, CF400-Cu) by scooping membrane fragments using tweezers. Two membranes for each synthesis condition (PA1, PA2 and PA3) were imaged using a JEOL 2100 Cryo TEM at an acceleration voltage of 200 kV with the electron dose maintained at rate of 4−7 e$^-$ Å$^{-2}$ s$^{-1}$ to ensure minimal to no beam-induced damage to the morphology. For electron tomographic reconstruction of the six membranes under study, 61 TEM images were collected every 2° by tilting the sample stage from 0° to −60° and 0° to

+60°, with a defocus value of −2048 nm and an objective aperture of 60 μm. The collected TEM images were aligned and contrast transfer function-corrected using IMOD 4.9.3[78], an open-source software by University of Colorado (http://bio3d.colorado.edu/). The tomograms for the PA membrane samples were generated by a Model-Based Iterative process using another open-source software from Purdue University called OpenMBIR[79] (https://engineering.purdue.edu/~bouman/OpenMBIR/). Volume 3D reconstructions, surface visualization and skeletonization of the PA membrane samples were performed using Amira 6.4 (Thermo Scientific).

## Morphology analysis

For the 3D reconstruction of the PA material volume, three filtering modules were applied to denoise the tomograms in the form of a median filter (26 neighbourhood, 26 iterations), Gaussian filter (kernel size 3, standard deviation $3 \times 3 \times 3$) and edge-preserving smoothing filter (time 5, step 5, contrast 3.5, sigma 3), followed by thresholding, semimanual segmentation and binary smoothing (kernel size $3 \times 3 \times 3$, threshold 0.5). The void reconstructions used the same filtering, thresholding and smoothing criteria as the material reconstructions, with an additional ambient occlusion module (rays 50, maximum distance) to identify the voids and the background. Once this module was applied, the PA material was used as a boundary to semi-manually segment the voids from the background of the tomogram. Skeletons of the PA material volume were generated using the autoskeleton module (smooth 0.5, iterations 10) with the skeleton thinning function in Amira 6.4 followed by extracting the statistics of the skeletons for further analysis. The branch lengths were normalized by the mean length across all 6 samples. The local thickness mapping was performed first by thresholding and binarizing the six tomograms using Amira 6.4 and then applying the local thickness plugin in FIJI ImageJ[34].

## AFM measurements

Elastic modulus mapping in air was performed using an Asylum Research Cypher S AFM. PA membranes were prepared for AFM analysis by scooping the PA membranes onto silicon wafer chips, cleaned with water, acetone and oxygen plasma treatment (1 min), followed by air drying. The Si wafer was mounted onto the sample stage in the microscope at ambient atmosphere. The model of the AFM tip used for tapping mode images, force curves and modulus mappings is AC240TS-R3 from Asylum Research Probes, with a typical spring constant of 2 N/m. Spring constant and the inverse optical lever sensitivity of the AFM tip were determined *via* thermal tune. Force mapping was performed in contact mode with a Z rate (vertical force spectra) of 100 Hz and a total pixel number of 256×256 points for each map. The height image, modulus map and force curves at different locations were obtained based on the force mapping measurements. Height images shown in Supplementary Fig. 1 were obtained using tapping mode images. The apparent modulus was calculated based on the fitting results using Hertz model with bottom-effect correction following the equation in our recent work[13]. We used sphere as the shape of tip, tip radius of 7 nm, tip Poisson of 0.17, tip modulus of 150 GPa and sample Poisson of 0.39 as the fitting parameters.

## Atomistic polymerization for MD simulations

To mimic the synthesis of PA films in the experiments, we placed the MPD and TMC monomers in distinct regions within a rectangular simulation box, emulating their immiscibility in solvents. The simulation box size is $L_x = L_z = 100$ nm with $L_y = 130$ nm. As shown in Fig. 3b, we employed a 2D graphene sheet that generated exclusively repulsive forces in the $xz$-planes of the box, where the monomers were situated, to maintain their separation. The graphene sheet, highlighted by an yellow line in Fig. 3b, is easily discernible.

The graphene sheet interacted with the benzene rings of the monomers through repulsive Weeks-Chandler-Anderson (WCA) potentials. Following a 20 ns equilibration period, we deactivated the repulsive interactions between the MPD monomers and the graphene sheet, allowing them to move freely and mix with the TMC monomers. The system featured a single interfacial region along the $y$-direction, leading to the formation of a single, independent membrane configuration, as depicted in Fig. 3c and Supplementary Fig. 8. Periodic boundary conditions were applied in three directions and two walls were positioned at the system's ends to prevent the leakage of the two monomer species. Additionally, vacuum conditions were set at the ends to prevent the interaction between the two monomer species due to the periodic boundary conditions.

As polymerization reactions progress and consume monomers, additional monomers diffuse toward the reaction zones due to concentration gradients. To keep the concentration in the bulk regions away from the interfaces constant, we designate specific regions within the cell as reservoirs with approximately constant monomer number density. As monomers diffuse from the reservoirs to the reaction zones, they are randomly replaced in the reservoirs at random sites and orientations.

In this work, two TMC:MPD monomer ratios per unit volume in reservoirs are considered, equal to 1:4 and 1:12.3 (TMC:MPD molar ratio), with MPD concentration of 5 w/v% used in aqueous solution in our experiments, higher than what has been commonly used in previous experiments[80]. The formation of clusters across the interface obstructed the diffusion of MPD into the TMC monomers region. These clusters grew gradually by reacting with monomers at any remaining surface sites and by incorporating diffusing clusters that were trapped between the forming clusters.

A modified version of the LAMMPS (Large-Scale Atomic/Molecular Massively Parallel Simulator) molecular dynamics package[43,81] is used to simulate the IP process. The developed CG force field[43] was employed to describe interatomic interactions, representing the component monomers, resulting oligomers and membranes. Lennard-Jones potentials are employed to characterise the non-bonding interactions between the CG monomers, because at long range monomers interact solely through Lennard-Jones potentials. To prevent any undesired overlap during IP, these potentials are augmented with repulsive interactions between side groups of the same kind to capture short-range interactions. In CGMD simulations, the crosslinking reaction is triggered when the distance between the carbon CG bead of the side group in the CG-TMC monomer and the nitrogen CG bead of the CG-MPD monomer falls below 2.375 Å, signifying the formation of a C-N (amide) bond within the simulation system. To ensure that each amine side group can only form a single amide bond, steric forces added in the simulation system is utilised to prevent two amide bonds for a single nitrogen atom. All simulations were conducted at 300 K and 1 atm using Langevin dynamics. The simulations ran for 600 ns, using a time step of 3.0 fs.

## CGMD simulation of the effect of monomer diffusion rates

The two CG monomer species (CG-MPD and CG-TMC) were initially segregated into distinct regions within the rectangular simulation box, as illustrated in Supplementary Fig. 12, mimicking the immiscibility of the solvent phases and phase separation. This separation was achieved using two 2D planes of exclusively repulsive interactions in the $xz$ planes across the simulation box. Similar to the simulation box for PA membrane formation reactions, these planes, represented by black dashed lines in Supplementary Fig. 12, act selectively with the benzene rings of the monomers through repulsive WCA potentials. Following a 10 ns equilibration period, the repulsive interactions between the black dashed lines and MPD monomers were deactivated. This allows MPD monomers to move beyond their initial confinement and enter the region where TMC remains confined. Due to the periodic boundary

conditions, two interfacial regions emerge in the system, resulting in the formation of two distinct membrane configurations.

## Permeance measurements

Permeance measurements for all membranes were conducted in methanol (HPLC-grade, Fisher Chemical) with an electronically controlled pressure-driven membrane testing cell (Sterlitech HP4750). The 25 mm diameter membranes were soaked in methanol for 24 hours prior to permeance testing. After equilibrium soaking, the membrane was transferred to the Sterlitech membrane cell and 100 mL of fresh methanol for permeance testing was added. The cell was slowly pressurized to 150 psig at a controlled rate of 25 psi/min with an electronically controlled gas manifold and the permeate flow rate was measured by recording the mass of permeated methanol over time. Once 80 mL of methanol permeate had been collected, the experimental run would end.

Membrane permeance was calculated from the average mass flow rate of permeate ($\dot{M}$), geometric membrane area ($A$) and pressure ($P$) with the following equation (Eq. (3)):

$$Permeance = \frac{\dot{M}}{A*P} \tag{3}$$

## Permeance modelling

Membrane permeance performance was analysed with a Spiegler–Kedem model to fit the methanol permeance of PA membranes with networked crumples[13]. First, a nominal membrane thickness was estimated with two different approaches. The detailed derivations of the two approaches are elaborated in Supplementary Note 4.

Approach 1 estimates the membrane nominal thickness using open and closed void surface area, $A_{OV}$ and $A_{CV}$, by assuming that open void surface areas are surrounded by $t_1$ thicknesses, the closed void surface areas are enclosed by one layer of $t_2$ thicknesses and another layer of $t_3$ thicknesses. The flat membrane region is of $t_2$ thicknesses. All the assumptions are from direct observations of tomographic reconstructions. Approach 2 uses membrane thickness from AFM measurements. Modelled permeance using each approach to estimate the nominal membrane thickness are compared in Fig. 4g.

## Chemical characterisation and DOC measurement

XPS (Kratos Axis) measurements were performed on PA membranes mounted on silicon wafers using an Al K-α X-ray (source energy of 1486.6 eV) and collected at a pass energy of 160 eV and at an energy step size of 1 eV. The DOC was calculated from the photoemission spectrum using a method described in literature[13,29] (Eqs. (4), (5)).

$$\frac{Os1}{Ns1} = \frac{3X + 4Y}{3X + 2Y} \tag{4}$$

$$DOC = \frac{X}{X + Y} * 100\% \tag{5}$$

Here, $Os1$ and $Ns1$ are integrated peak areas from the photoemission spectrum and X and Y are the number portion of crosslinked and linear structures.

## Characterising mechanical properties with MD simulation

We aimed to determine the experimental modulus of PA membranes with three different TMC:MPD ratios (0.01, 0.02 and 0.2) using IP simulations, as outlined in reference[71]. We constructed three corresponding IP atomistic models with an initial box size of $6.0 \times 6.0 \times 6.0$ nm$^3$ (Supplementary Fig. 18) and performed compression loading on each. To ensure a stable starting point, the initial model systems underwent a 21-step MD equilibration protocol[82]. After equilibration, we carried out a 2-ns simulation at 300 K and 0 MPa under the NPT ensemble. Compressive deformation was implemented using nonequilibrium molecular dynamics (NEMD) simulations, with the z-dimension of the simulation box (normal to interface between MPD and TMC) reduced at each simulation step to develop constant strain rate deformation in a $N\sigma_{ij}\varepsilon_{ij}T$ ensemble (where $N$, σ, ε and $T$ are particle number), stress, strain and temperature, respectively, up to a maximum engineering compressive strain of 0.2. The simulation system maintained zero pressure (0 MPa) along the x- and y-directions, allowing the dimensions normal to the loading direction to change in response to Poisson's effect, where PA material is extended horizontally as a result of vertical compression. The temperature was maintained at 300 K, the time step was 1.0 fs and the engineering strain rate was $10^8$ s$^{-1}$. We recorded the stress and strain at each step and plotted the stress-strain relationships for PA membranes with different TMC:MPD ratios, as shown in Supplementary Fig. 18. Finally, we extracted the corresponding modulus values for each membrane (Supplementary Fig. 18).

## Data availability

All data needed to evaluate the conclusions in the paper are present in the paper and/or the Supplementary Information. Correspondence and requests for additional data are available from the corresponding authors upon request.

## Code availability

Code availability: The open-source code used for the coarse-grained molecular dynamics simulations, LAMMPS, can be obtained from: https://figshare.com/articles/dataset/PolyamideFiles_zip/3803688. All input and output files from LAMMPS simulations are available upon request.

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

## Acknowledgements

We would like to thank the support by U.S. Department of Energy, Office of Basic Energy Sciences, under Award Number DE-SC0022035 (FCK, ET, CMS, JSM, QC) on the sample synthesis, preparation, electron tomography and other experimental characterization, the support by the Air Force Office of Scientific Research under award numbers FA9550-23-1-0609 and FA9550-20-1-0257 (FCK, JWS) on the tomography data analysis, and the support by National Science Foundation under award number 2243104 on GT analysis (LY). We acknowledge the use of resources at Central Research Facilities in Materials Research Laboratory and Beckman Institute for Advanced Science and Technology, University of Illinois at Urbana–Champaign.

## Author contributions

FCK, HA and QC conceptualized the work. FCK synthesized samples. FCK and HA performed electron tomography. Morphometry analysis was done by FCK (local thickness, void analysis) and JWS (local thickness). FCK constructed structural graphs and FCK and LY performed GT parameter calculations. MD simulations were performed by JH and YL. SC and XS performed permeance measurements and permeance modelling. AFM measurements were done by SZ (modulus mapping) and FCK (height maps, RMS calculations). The original draft sections for electron tomography, morphometry, GT (including Figs. 1, 2, 4, 5) were written by FCK and QC. The original draft sections for MD simulations (including Fig. 3) were written by JH and YL. All authors contributed to reviewing and editing the manuscript. QC, YL, XS, JSM, CMS and ET provided research advice. QC and YL supervised the work.

## Competing interests

The authors declare no competing interests.
