## [Peer Review File · Nature Communications]

Beyond Nothingness in the Formation and Functional Relevance of Voids in Polymer FilmsREVIEWER COMMENTS

Reviewer #1 (Remarks to the Author):

This is a very thorough investigation on polyamide (PA) active layer morphology using multiple advanced characterization and simulation approaches. It provides fundamental insights on the PA formation and the structural-performance relationship. The manuscript is clearly written with extensive analysis. I enjoy reading the manuscript and support its publication on Nature communication after minor revision.

I wonder how the CGMD captures the interfacial transport phenomenon of MPD and its impacts on the real stoichiometry in the reaction zone (likely in the oil phase). The authors also mentioned a few explanations regarding void formation, including reaction heat and interfacial boiling. Clearly MD does not seem to be able to capture (or disprove the presence of) those effect. How does CGMD then can predict the key morphological features the PA layer?

For MD simulations, the authors should read the papers from Richard Lueptow, Ying Li, and others and compare the current approach to theirs. This recent paper (<https://doi.org/10.1016/j.memsci.2023.121980>) also provides important insights on the stoichiometric effect in the reaction zone. Early work by Freger on PA active layer formation should also be discussed and compared against.

Reviewer #2 (Remarks to the Author):

This work investigates the nanomorphology, the formation mechanism and the corresponding solvent permeance and mechanical strength of the typical IP-formed polyamide membranes with interconnected nanovoids, via 3D imaging, morphometry platform and coarse-grained molecular dynamics (CGMD) simulation for the first time. Some interesting results and novel models are presented and discussed. However, for a comprehensive research study, corresponding experimental results should be provided, to explain to nanomorphology variation of the membranes, support the simulation result, and for a better understanding of the authors. A major revision is recommended as noted below.

1. In addition to the typical composite PA membranes, freestanding PA membranes following the same IP conditions (without the substrate) should be fabricated for convenient experimental characterizations, such as XPS, density, and DMA test, to demonstrate the consistency to the membrane nonomorphology.
2. The local thickness and void size of the PA membranes are given by the tomography-based imaging method. The authors should also provide an explanation or chemical characterizations to explain to trend of PA membranes fabricated under different conditions.
3. Similarly, CGMD simulations show that "...the membrane formation process can be described by a combination of self-limiting reaction-aggregation process and reactant diffusion." And, "..... a clustered network is formed by coalescence of different oligomers, which continues to grow as additional

monomers and/or oligomers accrete at reaction sites on the membrane surfaces (Fig. 2f) until the reactants are depleted or the membrane becomes continuous that self-limits the growth.” It is more convincing if the authors can provide some experimental proof for the statement. Probably the diffusion rate of CG-MPD / MPD monomers? Or monomers with different diffusion rates as a supplementary proof?

4. Fig. 3 (c) demonstrates a 3D nanovoid map reconstruction for PA membranes showing open voids and closed voids. So how about the effect of such open and closed voids in PA membranes on the solvent permeance of membranes? Can the volume/ratio of such open and closed voids be linked with the membrane properties?

5. Line 298, this paragraph shows that “the concentration of TMC monomer is shown to serve as a handle to tune the morphology”. The authors should link the void space percentage of the PA membrane (maybe the freestanding PA membranes) with their physical properties (such as density and mechanical properties) and solvent permeance.

6. The authors characterize the membrane mechanical strength by AFM. Have the authors tried to characterize the freestanding PA membrane by DMA? And Fig. 4h shows the apparent moduli increases as cTMC increases, which is in an opposite trend of the degree of crosslinking (Supplementary Fig. 10b). The author may justify this. Can this trend be linked with the nanomorphology variation?

Reviewer #3 (Remarks to the Author):

The submitted work uses novel TEM tomography analysis and data processing to study the structure of interfacially polymerized polyamide films and how this structure relates to solvent permeance. Specifically, the study focuses on characterization of the nanovoid structure of these films, a structure that has become apparent in the past ~8 years and that is important to understand to advance the elucidation of property-performance relationships of polyamide membranes. The study uses novel approaches, provides novel insights, is interesting, and I believe will ultimately be an important addition to the literature on polyamide membranes. Before publication I recommend the authors address the comments below.

-One point that stood out to me was that this being a paper about nanovoids in interfacially polymerized polyamide films, there was no discussion on how the observations by the authors validate or are in conflict with the prevalent theory on how these nanovoids originate. There is relatively extensive experimental work demonstrating that nanobubbles created during the reaction of MPD and TMC give origin to the nanovoids. Much of this work has been led by the Tang laboratory in Hong Kong University (e.g., <https://pubs.acs.org/doi/10.1021/acs.est.9b05892>). Others have used this theory to rationalize novel polyamide film structures (e.g., very large voids, multi-layered void structures) obtained with interfacial polymerization conditions different from typical conditions (<https://doi.org/10.1016/j.memsci.2021.120112>). Given that the authors have performed novel analysis of physical and in silico interfacially polymerized polyamide films, do the authors find their results consistent with this prevalent theory about nanobubble induced void formation? Tang et al

demonstrated that by eliminating nanobubble formation, they also eliminated nanovoids. However, the authors do not account for bubble formation in their coarse grain polymerization model and still seemingly obtain voids in the polymer structure? How do the authors reconcile these two results? Overall, a discussion of how the submitted work is consistent or not (and why) with the nanobubble theory of void formation is important.

-Another point that stood out to me is that throughout the paper the authors do not compare any of the values they obtain from the analysis of their physical or in silico membranes (i.e., permeance, thickness, void fraction, etc.) or membrane fabrication values (i.e., MPD and TMC concentrations and ratios) with those used typically for production of well-performing polyamide membranes in the industry or literature. Such comparisons are important to put in perspective the results and conclusions reported by the authors. For example, were the concentrations/ratios used to prepare the membranes consistent with what is typical for well-performing membranes? If not, please specify why the concentrations/ratios used in this study were chosen? How does the permeance, thickness, void fraction, crosslinking, etc., of the physical and in silico membranes compare with values in the literature for commercial or lab-made membranes?

-The authors say vaguely (pages 15-16) how they 'predicted' solvent permeance; they indicated they used the Spiegler-Kedem model or some modification of it. However, they do not explain in the Results, Methods, or Supplementary Materials what transport equations they used for this, what the inputs were to the equations, etc. They only provide equations about thickness (eqs 2, 3) and other basic coefficient equations about coefficients in transport models (eqs 4-6). These, however, are not sufficient to understand how the transport was modeled. For example, the authors state that they used the degree of crosslinking in the modeling but none of the equations presented have crosslinking as one of the parameters. It is important the authors explain the transport modeling clearly (even if in the Supplementary Materials) since results from that modeling are an important part of Figure 3 and corresponding discussion.

-It is unclear what the purpose of the comparison between 'model estimates' and experimental measures of permeance is. The authors use the word "predict/prediction" for the model estimates but the "permeability" constant used in the model estimates of permeance (i.e., μ in eqs 5, 6) was calculated directly from the experimental data (permeability, thickness, and density). To calculate the model estimate, the authors simply used the same equation (eq 6) used to analyze the experimental data (and the permeability extracted from it using the experimental data) but now with the densities and thicknesses obtained from the polyamide layers generated in silico. Therefore, the model estimates are not a true prediction of permeance (i.e., because the calculations used the permeability obtained from experiment). The only purpose I can see for comparing 'model estimates' and experimental measures of permeance is to evaluate whether the product of thickness and density for one case (e.g., in silico) is consistent or not with the other case (e.g., experiment).

-The definition of local thickness (L159-160) is not very clear. Does 'local thickness' refer to that of a film of continuous polyamide or to a descriptive thickness at a specific position in the x-y plane? For example, in Figure 3b(ii) it is clear that in some instances when moving along the z-axis (i.e., perpendicular to the x-y plane) one traverses polyamide, then void, then polyamide again (e.g., right hand side of Fig 3b(ii)). Would there be two local thicknesses at that x,y location? or just one? If one, what would be the thickness at that x,y location? The summation of the diameter of the sphere that fits into the top polyamide film plus the diameter of the sphere that fits into the bottom polyamide film? Please clarify. My understanding is that the thickness mappings in figures and videos are associated with x,y locations.

-I am not convinced the important statement in L172-L175 that $t_2 \sim 2t_1$ and $t_3 \sim 3t_1$ is accurate. From Fig 2 it appears that $t_1 \sim 5\text{nm}$, $t_2 \sim 15\text{nm}$ and $t_3 \sim 20\text{nm}$. Therefore $t_2 \sim 3t_1$ and $t_3 \sim 4t_1$. This is an important detail because then the reasoning for the origin of the thicker regions may require some revision. Can the authors tabulate these thicknesses?

-The main use of polyamide membranes in real life (and in academic literature) is water treatment. Therefore it seems strange that the authors evaluated methanol permeance instead of water permeance for this study. I am not aware of the use of polyamide membranes for methanol filtration in industry as an established practice (though perhaps I may just be unaware of it). The paper would be stronger if it evaluated water permeance, though if that is not possible at the moment, providing a reason to evaluate methanol permeance instead of water permeance may be informative to the reader.

OTHER COMMENTS

-L152-L154: it is unclear what the reference to Geise's work is for. Do the authors mean that Geise et al did not have access to accurate thickness values and therefore their prediction of the upper bound could be improved in the future with more accurate thickness measurements like the ones in the submitted work? or did the authors simply mean to give an example of when accurate thickness values may be needed?

-L218-L220: I do not understand the mechanism the authors try to explain in one sentence here. They mention "this instability" but they do not mention any instability in the preceding sentences.

-The statement in L317 "that monomer concentration alone is also a good strategy to increase void volume" is supported by the results in a recent relevant study (<https://doi.org/10.1016/j.memsci.2021.120112>) that demonstrated experimentally that the MPD concentration and availability affects polyamide film thickness and void structure and fraction.

-Fig 1a: The authors state in the caption that "The schematic is zoomed-in to show the interconnected crumple morphology" but that interconnected crumple morphology is not clear. Making bigger the inset may suffice.

-Fig 2a: the left panel (especially what the spheres correspond to) is not very clear.

-Fig. 3d: what do the colors represent?

-Equations 2 and 3: what is the conceptual basis for these equations. Please add to the Supplementary Materials.

-Supplementary Fig 4c: it is unclear what this figure shows, especially the right panel of the image.

Responses to Reviewer 1

Reviewer 1 commented positively on our work as a “*very thorough investigation...using multiple advanced characterization and simulation approaches*” and supported “*its publication on Nature communication after minor revision*”. We thank the reviewer for their insightful comments, all of which are addressed with new simulations (new Fig. 3, Supplementary Figs. 11,12) and discussions (Supplementary Fig. 10, new Supplementary Notes 2,3) as detailed below.

Comment 1: *I wonder how CGMD captures the interfacial transport phenomenon of MPD and its impacts on the real stoichiometry in the reaction zone (likely in the oil phase).*

Reply: We thank the reviewer for the great comment! The CGMD model captures the interfacial transport of MPD monomers by considering the full diffusion-reaction process of interfacial polymerization (IP), so as to simulate the conformational bonding of monomers, their aggregation into PA oligomer clusters, and the morphology development into the PA membrane. The stoichiometry in the reaction zone is controlled by the diffusion-reaction process, which we can also vary by changing the diffusion rate. Note that since we used implicit solvents to overcome challenges in computation time, the absolute concentrations of monomers are higher (in the absence of explicit solvents) than what was used in experiments. Nevertheless, in our simulation, we kept the TMC:MPD monomer concentration ratios the same as in our experiments to study the impact of concentration effects.

In the revised manuscript, we have made the following revisions to address the comment in detail:

1. Added new discussions in the main text and a new Figure 3 to detail how CGMD captures the interfacial transport phenomena:

“For the CGMD simulations, the monomers are coarse-grained into representations, where the bond lengths, molecular conformations as well as reactivity are consistent with those of a real atomistic structure. The reacting MPD and TMC monomers are represented as a triad of beads arranged in a rigid equilateral triangle with pendant reactive groups (Fig 3a).”

Fig. 3. CGMD simulations on the mechanism of nanovoid and crumple formation. (a) Depictions of (i) atomistic and (ii) CG structures for TMC monomer, MPD monomer, and reacted TMC/MPD dimer. Note that carbon, nitrogen, oxygen, and hydrogen atoms are shown as grey, blue, light green and white circles, respectively. (b) Simulation setup showing the initial distribution of TMC (purple) and MPD (green) monomers within the simulation cell, with the reservoir boundaries depicted in red dashed lines. To prevent

the TMC and MPD monomers from coming into proximity, a repulsive potential represented by a yellow line is employed. To prevent their interaction due to periodic boundary conditions, two 25.0 nm vacuum layers are incorporated on two sides. (c) Based on a TMC:MPD monomer ratio of 1:12.3, the snapshots showcase the formation of a spanning membrane in a CGMD simulation at three successive times: 0 ns, 90 ns, and 600 ns. These snapshots are arranged from left to right, providing a visual depiction of the progressive stages of membrane formation over time. Best perspective views are selected to show the oligomer and spanning membrane distributions. The simulation box ($100 \times 130 \times 100$) nm³ at 0 ns shows the initial distribution of TMC (purple) and MPD (green) monomers before permitting IP reactions. (d) Zoomed-in region of the spanning membrane showing nanovoids (shown as grey regions within black dashed lines). Scale bar: (d) 4 nm.

“Next, we set up the computational cell such that the initial distribution of CG-TMC and CG-MPD is phase-separated (gridded) following experimental conditions, and IP starts as we switch off the repulsion between the grids and CG-MPD monomers, allowing them to diffuse, mix with CG-TMC, and form into PA membranes (Fig. 3b). By varying ratios of CG-MPD and CG-TMC monomers in the reservoirs, we can achieve the desired stoichiometry during the IP process. Thus, this approach facilitates simulations that capture the essential stereochemical features of the monomers and the amide bonding between them.”

- Added Supplementary Note 3 and Supplementary Fig. 11 based on new simulations on the impact of monomer diffusion rate on the stoichiometry in the reaction zone and the final PA morphology.

In the main text, we added,

“We also incorporate additional simulations to investigate the impact of amine monomer diffusion rates on the IP process, where denser membrane structures are formed when the diffusion rate of the amine monomer is low (Supplementary Note 3, Supplementary Fig. 11,12).”

Supplementary Fig. 11. Effect of monomer diffusion rate on PA morphology characterized by CGMD.

For a starting monomer ratio of 1:1 TMC:MPD, snapshots illustrate the evolution of a spanning membrane in a CGMD simulation in both (A) *yz* plane and (B) *xy* planes at four consecutive time points: 0 ns, 10 ns, 60 ns, and 120 ns. The molecular mass of MPD was varied to simulate three distinct rates of diffusion, as 0.3 times the original monomer mass (top), the original monomer mass (middle), and 3 times the original

monomer mass (bottom). The snapshots are sequentially arranged from left to right, providing a visual representation of the successive stages of membrane formation over time. Scale bars: 5 nm.

Supplementary Note 3: Impact of monomer diffusion rates on PA membrane morphology

“We incorporated three additional simulations to investigate the impact of MPD diffusion rates on stoichiometry within the reaction zone and the final PA membrane density. To optimise computational efficiency, we utilised the model and simulation setup previously established by Muscatello *et al.* (*J. of Membr. Sci.* **527**, 180–190 (2017)). The simulation box dimensions are $(10 \times 10 \times 200)$ nm³, with a TMC:MPD concentration ratio of 1:1. Subsequently, to characterise three distinct monomer diffusion rates, we arbitrarily varied the mass input for the MPD monomer as 0.3, 1.0, and 3.0 times the original molecular mass (*Phys. Rev. Lett.* **129**, 048003 (2022)). Note that the diffusion rate of MPD monomers in hexane is 10^{-6} cm² s⁻¹ (*Science* **360**, 518–521 (2018)), while in our CGMD simulations, the diffusion rate of CG-MPD monomers is 1.265×10^{-6} cm² s⁻¹, a result of using implicit solvent conditions. We observed that diverse diffusion rates affect the local membrane configuration during the reaction process, resulting in denser membrane structures when the rate of diffusion of MPD monomer is low and less dense membrane structures with a higher rate of diffusion compared to the original monomer diffusion rate. In other words, the different diffusion rates of the amine monomers affect the monomer stoichiometry in the reaction zone as observed with the distinct local membrane configuration (Supplementary Figs. 11,12).”

Added descriptions in the “Methods” section and Supplementary Fig. 12 on how the new CGMD simulations are set up:

“The two CG monomer species (CG-MPD and CG-TMC) were initially segregated into distinct regions within the rectangular simulation box, as illustrated in Supplementary Fig. 12, mimicking the immiscibility of the solvent phases and phase separation. This separation was achieved using two 2D planes of exclusively repulsive interactions in the *xz* planes across the simulation box. Similar to the simulation box for PA membrane formation reactions, these planes, represented by black dashed lines in Supplementary Fig. 12, act selectively with the benzene rings of the monomers through repulsive WCA potentials. Following a 10 ns equilibration period, the repulsive interactions between the black dashed lines and MPD monomers were deactivated. This allows MPD monomers to move beyond their initial confinement and enter the region where TMC remains confined. Due to the periodic boundary conditions, two interfacial regions emerge in the system, resulting in the formation of two distinct membrane configurations.”

Supplementary Fig. 12. CGMD setup for diffusion rate simulations. (A) The initial distribution depicts TMC (light green) and MPD (orange) monomers within the simulation cell before initiating interfacial polymerisation reactions. Grids of repulsive potentials, denoted by black dashed lines, were employed to maintain separation between TMC and MPD monomers. (B) After the initial equilibration, the repulsive interactions between the black dashed lines and MPD monomers were disabled. This allows MPD monomers to diffuse from the orange region, where they were initially confined, into the light green region. TMC monomers remain confined to their original region, as shown in (A). The locations of reservoirs for MPD and TMC monomers are outlined by thin black dashed lines. Polymerisation reactions occur when MPD comes into contact with TMC. Periodic boundary conditions were applied to all faces of the simulation cell.

Comment 2: *The authors also mentioned a few explanations regarding void formation, including reaction heat and interfacial boiling. Clearly MD does not seem to be able to capture (or disprove the presence of) those effect. How does CGMD then can predict the key morphological features of the PA layer?*

Reply: We appreciate the reviewer for highlighting this important point. Indeed, in our CGMD simulations, the system is maintained at a relatively constant temperature under Langevin dynamics, which is limited in modeling local effects such as reaction heat and interfacial boiling. We instead focus on using the CGMD simulation to capture the diffusion-reaction of monomers and the IP process, which shows the formation of polymer oligomers and PA membranes that are *interconnected and contain voids*, consistent with our experimentally resolved morphologies. The simulation also allows us to elucidate the mechanism of “oligomer coalescence-and-growth”, which matches with the local thickness histograms that we measured from electron tomography. This match between the CGMD simulation and experiments highlights that the monomer diffusion-reaction characteristics determine the key morphological features of the PA membrane.

In the revised manuscript, we have made the following revisions to detail the CGMD simulation and the morphologies produced by the model.

1. Added discussions in the main text on the model:

“Note that in our CGMD simulations, the system is maintained at a relatively constant temperature under Langevin dynamics, which is limited in modelling local effects such as reaction heat and interfacial boiling. Nevertheless, the match between the PA membrane morphologies from the CGMD simulation and our experiments highlights that the monomer diffusion-reaction characteristics and the resulting “oligomer coalescence-and-growth” mechanism determine the key morphological features of the PA membrane.”

2. Added more detailed discussions in the main text and new figures, including Fig. 3d (see our response to Reviewer 1, Comment 1) and Supplementary Fig. 10, on the morphology details of voids and molecular mechanism of the PA membrane generated from the CGMD simulation:

“In the CGMD simulation, we observe that the PA membrane shapes through the formation and coalescence of oligomer clusters at the interface with three key structural features. Firstly, while the simulation maintains a constant MPD:TMC ratio, a single CG-MPD monomer yields two amine groups, whereas one CG-TMC monomer contributes three carboxyl groups. Consequently, there is an imbalance in the number of reactive groups of CG-MPD and CG-TMC. Secondly, the PA membrane consists of regions where the degree of polymerization is high, interspersed with interfaces where the degree of polymerization is relatively low. Lastly, significant variations in local density are observed, including voids. These structural features captured in the CGMD simulations align with our experimental results and previous references using ensemble spectroscopy methods (*Environ. Sci. Technol.* **39**, 1764–1770 (2005)).

“The CGMD simulation also allows us to track the spatial variations of surface chemistry during the IP process. Interestingly, we find that prior to the coalescence of oligomer clusters, unreacted acyl chloride and amine side groups tend to reside predominantly on the surfaces of the clusters.

Once two clusters coalesce, it becomes challenging for the unreacted side groups on the cluster surfaces to undergo further reactions due to steric hindrance or an imbalance in the number of amine and carboxyl reactive groups. As a result, the interfaces between oligomer clusters can have sites that are not reacted or forming into amide bonds, leading to the formation of voids in the PA membranes (Fig. 3d, Supplementary Fig. 10).”

Supplementary Fig. 10. Nanovoids visualised in CGMD simulations. Snapshots of CGMD simulations of PA membranes with TMC:MPD monomer ratios of (A) 1:4 (PA4) and (B) 1:12.3 (PA3) after 600 ns. Snapshots of PA membrane with TMC (purple) and MPD (green) monomers (left), PA membrane without monomers to show the PA oligomer clusters (middle), and a zoomed-in view of the nanovoids (right). Note that the nanovoids are outlined using dashed lines. The box dimensions are $(100 \times 130 \times 100) \text{ nm}^3$. Scale bars: 4 nm.

Comment 3: For MD simulations, the authors should read the papers from Richard Lueptow, Ying Li, and others and compare the current approach to theirs. This recent paper (<https://www.sciencedirect.com/science/article/pii/S0376738823006361?via%3Dihub>) also provides important insights on the stoichiometric effect in the reaction zone. Early work by Freger on PA active layer formation should also be discussed and compared against.

Reply: Thank you! In the revised manuscript, we added discussions and more references on comparisons of our CGMD simulation with previous work in both the main text and SI text, with the latter more detailed.

1. Added the discussions in the main text:

“It is worth noting that there are various simulation approaches available to describe the formation of the IP process of the PA membrane, mostly based on atomistic simulations (details in Supplementary Note 2). While PA membranes constructed at the atomistic-level offer a close representation and prediction of local characteristics such as dry and hydrated densities, and molecular transport properties, these models are computationally expensive, often used for computational cell sizes of $\sim 5\text{--}20 \text{ nm}$ in the x and z dimensions (*Polymer* **55**, 1420–1426 (2014), *J. of Membr. Sci.* **686**, 121980 (2023)). As a result, these simulations cannot accurately reproduce experimentally observed membrane morphologies, which can have voids and structures on the scale of 100s of nm and require longer timescales to form. In our work, we use the CGMD model developed by Muscatello *et al.* (*J. of Membr. Sci.* **527**, 180–190 (2017)) to overcome the limitations. This CGMD approach utilises a multi-scale modelling technique, where the CG models maintain the shape and connectivity of the monomers, while simulating monomer diffusion without explicitly modelling the solvents themselves. Additionally, it offers the ability to map these CG

models onto fully atomic configurations through a subsequent relaxation procedure. By employing this multi-scale modelling technique, we can achieve accurate representation of the monomer dynamics and bond formation while still benefiting from the computational efficiency provided by CGMD modelling.”

2. Added a new Supplementary Note 2 with more detailed discussions.

Supplementary Note 2: Previous work on molecular dynamics simulations of PA membranes

“Previous simulation efforts on PA membrane formation are mostly based on atomistic models. In a study conducted by Kolev *et al.* (*Polymer* **55**, 1420–1426 (2014)), atomistic PA membranes were generated to provide insights into the mechanisms involved in the formation, hydration and functioning of membranes during the interfacial polymerization (IP) processes. To simulate this, they initially set up a simulation box containing MPD and TMC monomers, along with a few TMC/MPD dimers acting as initial clusters. Following the initial setup, the monomers in the simulation were allowed to react exclusively with the growing clusters, while preventing reactions between monomers. With the formation of each new bond, a monomer of the same type as the one that reacted was introduced into the simulation box. This introduction occurred randomly within the vacant space not occupied by the van der Waals volume of the polymeric clusters. This simulation procedure closely resembles the stoichiometrically balanced diffusion of monomers from adjacent solutions into the reaction zone, as observed in IP processes, although the size of the simulation box is too small to capture nanomorphology.

Li *et al.* (*J. of Membr. Sci.* **686**, 121980 (2023)) performed equilibrium molecular dynamics (EMD) simulations to simulate the construction process of PA membrane at an atomic level. They investigated the morphological transformation and mass transport through the polymer network by varying the stoichiometry of the monomers. In their study, MPD and TMC molecules were initially placed in a cubic cell using a heuristic algorithm to ensure randomization. To achieve crosslinking between MPD and TMC, they utilised an update to the reaction radius. Specifically, if the nitrogen atom in a free amine group came within 3.25 Å of a carbonyl carbon atom in a free acyl chloride group, a molecular topology transformation was triggered. This transformation involved the formation of an amide bond while simultaneously removing any excess hydrogen and chlorine atoms present. The crosslinking process involved continuous updates to the atomic radii of the reaction until reaching either the maximum C-N cutoff distance or the desired conversion target. Different MPD:TMC ratios were considered, corresponding to different stoichiometric ratios based on the amine:acyl chloride functional group molar ratio.

He *et al.* (*Desalination* **547**, 116204 (2023)) performed non-equilibrium molecular dynamics (NEMD) simulations to investigate the effect of different manufacturing methods on the performance of PA membranes. For IP in particular, a fixed 3:2 MPD:TMC ratio was used for the simulations. The reacted atoms were identified and arranged into TMC and MPD monomers. Each monomer type provided specific potential reaction sites. These monomers were then filled into separate 3D-periodic cells of equal cross-section size. The two cells were assembled along the direction normal to the cross-section, creating a composite with an interfacial layer between the MPD and TMC layers. The cross-linking reaction occurred randomly and was restrained to the MPD/TMC interface, simulating the IP process.

In a study by Shen *et al.* (*J. of Membr. Sci.* **506**, 95–108 (2016)), NEMD simulations were used to study the atomic-scale transport of water, ions, and small organic solutes in a commonly used membrane. In this process, the TMC and MPD monomers were randomly moved in a computational box. When the functional groups of the monomers responsible for cross-linking were within a specified distance from each other, an amide bond was formed, subsequently building the polymeric structure. The aim was to mimic the variability observed in actual polymerization processes, considering the inherent randomness involved.”

Responses to Reviewer 2

Reviewer 2 commented that “*for the first time,*” “*this work investigates the nanomorphology, the formation mechanism and the corresponding solvent permeance and mechanical strength of the typical IP-formed polyamide membranes with interconnected nanovoids, via 3D imaging, morphometry platform and coarse-grained molecular dynamics (CGMD) simulation,*” which reveals “*interesting results and novel models.*” The reviewer recommended “*a major revision*” to include additional experimental results. We thank the reviewer for their comments to make our study more comprehensive, all of which we have now addressed with new experiments (Supplementary Figs. 3,7, Supplementary Table 3), simulations (Supplementary Fig. 11,12), and discussions (Supplementary Notes 1,3) as detailed below.

Comment 1: *In addition to the typical composite PA membranes, freestanding PA membranes following the same IP conditions (without the substrate) should be fabricated for convenient experimental characterizations, such as XPS, density, and DMA test, to demonstrate the consistency to the membrane nanomorphology.*

Reply: We thank the reviewer for the great comment about studying how the presence of substrate during the IP process affects the chemical properties and nanomorphology of PA membranes. In the revision, we have synthesized freestanding PA membranes without the polysulfone substrate at the same IP conditions (hereafter referred to as “*control PA membranes*”) and performed XPS to characterize their degree of crosslinking and elemental composition) and electron tomography to resolve their 3D nanomorphology. In summary, the control membranes are similar to those synthesized in the presence of substrate (referred to PA membranes unless otherwise noted, which is the focus of study in our manuscript), in terms of composition, degree of crosslinking and morphologies with interconnected crumples and voids (Supplementary Fig. 3). This consistency supports the formation mechanism revealed by the CGMD simulations, where the substrate is not considered.

In the revised manuscript, we have made the following changes to address the comments:

1. Performed new experiments of synthesizing and characterizing the control PA membranes (without the polysulfone supporting substrate),

Added new discussions in the main text:

“Furthermore as a comparison, control PA membranes are synthesised without the use of a PS support layer following the same starting concentrations and reaction times, to demonstrate the consistency of the observed interconnected and nanovoid-containing morphology (Methods, Supplementary Note 1, Supplementary Fig. 3).”

Added new Supplementary Fig. 3 and Supplementary Note 1 on the imaging and reconstruction of the control PA membranes:

Supplementary Note 1: Synthesis and nanomorphology characterization of control PA membranes

“Control PA membranes were synthesised without the supporting PS substrate layer, with similar starting monomer concentrations to the PA membranes under study. Thus, control PA1, control PA2 and control PA3 were prepared with 0.05, 0.1 and 0.1 w/v% c_{TMC} and a constant c_{MPD} of 5 w/v%. As evident from the two-dimensional (2D) TEM and three-dimensional (3D) electron tomographic images (Supplementary Fig. 3), the control PA membranes show an interconnected crumpled morphology, with distinct crumple densities and crumple sizes for each starting monomer condition. Similar to their counterparts synthesised in the presence of PS substrate (i.e. PA1, PA2 and PA3), the crumple density of the control membranes increases as c_{TMC} increases. The crumple walls of the control membranes are interconnected and the membranes have nanovoids spanning the 3D space (Supplementary Fig. 3). The degree of crosslinking (DOC) and elemental composition of the control PA membranes are comparable to those of the PA membranes under study, with DOC

following the same trend (control PA1 > control PA2 > control PA3) (Supplementary Fig. 3, Supplementary Table 3).”

Supplementary Fig. 3. TEM imaging and 3D tomographic reconstructions of control PA membranes synthesised without supporting substrates. (A) TEM images of control PA1, control PA2 and control PA3 membranes synthesised without the PS support layer. Note that the starting monomer concentrations for control PA1, control PA2 and control PA3 are $c_{TMC} = 0.05, 0.1$ and 1 w/v%, respectively, and $c_{MPD} = 5$ w/v% for all. (B) Grayscale intensity-based electron tomographic reconstructions of control PA membranes showing an interconnected crumpled morphology similar to their counterparts synthesised in the presence of PS support layer. The projection area of reconstruction is $727 \text{ nm} \times 727 \text{ nm}$, and the reconstruction height varies with the crumple heights. (C) The bottom xy -slices of $727 \text{ nm} \times 727 \text{ nm}$ for the three membranes showing inner void regions and interconnected crumple walls. Scale bars: 200 nm .

2. Characterised the degree of crosslinking and elemental composition of the control PA membranes to compare them with the PA membranes, and revised Supplementary Fig. 7 and Supplementary Table 3 as follows:

Supplementary Fig. 7. Characterisation of crosslinking density of PA membranes. XPS spectra of PA and DOC presented as a function of monomer concentration ratios for (A) PA membranes under study and (B) control PA membranes as a comparison.

Supplementary Table 3. Atomic composition and DOC generated from XPS measurements for PA membranes synthesised with substrate and the control PA membranes synthesised without substrate (Supplementary Fig. 7).

	Atomic composition from XPS (%)			DOC (%)
	C	O	N	
PA1	74.80	12.70	12.50	97.7
PA2	75.51	13.31	11.18	73.9
PA3	76.58	13.98	9.44	41.8
Control PA1	76.91	12.05	11.04	86.88
Control PA2	77.54	12.56	9.90	64.47
Control PA3	78.83	12.96	8.21	32.69

3. Added details in the “Methods” section on the synthesis procedure of the control PA membranes:

“The synthesis of control PA membranes (i.e. PA membranes without supporting PS layer) was performed using a method described by Cui *et al.* (*Ind. Eng. Chem. Res.* **56**, 513–523 (2017)). Control PA membranes were synthesised by carefully adding TMC in hexane, with concentration depending on the desired control PA membrane starting concentration (10 mL, 0.05w/v% for control PA1, 0.1w/v% for control PA2 and 1 w/v% for control PA3) along the wall of a glass petri dish containing an aqueous solution of MPD (10 mL, 5w/v%). The control PA membrane was formed at the hexane–water interface. After 1 min, the reaction was stopped by carefully pipetting out the TMC in hexane solution (without damaging the fragile PA membrane) and washing three times with hexane (10 mL×3). The excess hexane was removed, and the membrane was transferred to float on the air–water interface of a petri dish with water.”

4. Added details in the “Methods” and revised Supplementary Fig. 1 to detail the sample preparation procedure of the PA membranes synthesized in the presence of PS support, to make it clear that those membranes were synthesized as composite membranes but were characterized in their freestanding forms following the literature conventions (*Science* **348**, 1347–1351 (2015), *Sci. Adv.* **8**, eabk1888 (2022)) for electron tomography:

“With the support of the PS support layer, the synthesised PA membrane along with the Cd(OH)₂ NS layer was carefully transferred to float on the air-water interface of a Milli-Q water-filled petri dish, resulting in a free-standing membrane.”

Supplementary Fig. 1. TEM images and AFM height maps of PA membranes with interconnected voids and crumples synthesised at three different starting monomer concentrations (c_{TMC} , c_{MPDI}). (A) Photograph of a freestanding PA membrane floating on the air–water interface. The membrane is outlined using white dashed lines. (B) TEM micrographs of PA1, PA2 and PA3 at two different magnifications show the morphology differences between the three systems. (C) AFM height maps of PA1, PA2 and PA3 show the membrane height variations.

Comment 2: *The local thickness and void size of the PA membranes are given by the tomography-based imaging method. The authors should also provide an explanation or chemical characterizations to explain to trend of PA membranes fabricated under different conditions.*

Reply: Thank you! To address this comment, in the revised manuscript, we have added more discussions to explain the trend of local thickness and void size in the PA membranes with varying monomer ratios.

1. We added discussion in the main text relating the local thickness to the degree of crosslinking of the PA membranes synthesized at different conditions:

“The observed decrease in local thickness with the increase of c_{TMC} can be explained by decreases in the degree of crosslinking (DOC) of the PA membranes (Supplementary Fig. 7, Supplementary Table 3). Previous studies showed that the PA membranes with higher DOC are thicker than those with lower DOC (*Desalination* **287**, 310–316 (2012)).”

2. We revised the main text to emphasize the influence of monomer concentration and reaction-diffusion instability on the void size of the PA membranes:

“PA1 has large, clustered nanovoids spanning through the reconstructed volume compared to PA2 and PA3. As discussed in Fig. 1c, the density of crumples is governed by a reaction-diffusion instability, resulting in higher crumple density at high c_{TMC} , and vice versa. PA1 has the lowest c_{TMC} , hence the largest separation between crumples, resulting in the lowest crumple density of the three membranes. Therefore, during membrane formation, the reaction front in PA1 deforms into a larger region compared to the crumples of PA2 and PA3, leading to larger voids. PA3 on the other hand, has the highest c_{TMC} and the highest crumple density among the three membranes, resulting in smaller crumples and smaller voids.”

Comment 3: Similarly, CGMD simulations show that “...the membrane formation process can be described by a combination of self-limiting reaction-aggregation process and reactant diffusion.” And, “..... a clustered network is formed by coalescence of different oligomers, which continues to grow as additional monomers and/or oligomers accrete at reaction sites on the membrane surfaces (Fig. 2f) until the reactants are depleted or the membrane becomes continuous that self-limits the growth.” It is more convincing if the authors can provide some experimental proof for the statement. Probably the diffusion rate of CG-MPD / MPD monomers? Or monomers with different diffusion rates as a supplementary proof?

Reply: We thank the reviewer for the great comment on the effects of monomer diffusion rates. It has been experimentally measured that the diffusion rate of MPD monomers in hexane is $10^{-6} \text{ cm}^2 \text{ s}^{-1}$ (*Science* **360**, 518–521 (2018)), while in our CGMD simulations, the diffusion rate of CG-MPD monomers is $1.265 \times 10^{-6} \text{ cm}^2 \text{ s}^{-1}$. This slight discrepancy between the experimental and simulated rates of diffusion is due to the use of implicit solvent molecules. In the revision, as suggested by the Reviewer, we performed new simulations using amine monomers of three distinct diffusion rates, and found that the rate of diffusion affects the density of the membrane. These results are consistent with previous literature (*Nat. Commun.* **13**, 500 (2022), *Macromol. Mater. Eng.* **306**, 2000818 (2021)). Please refer to our response to Comment 1, Reviewer 1, for all the changes we have made in addressing the diffusion rate question.

Comment 4: Fig. 3 (c) demonstrates a 3D nanovoid map reconstruction for PA membranes showing open voids and closed voids. So how about the effect of such open and closed voids in PA membranes on the solvent permeance of membranes? Can the volume/ratio of such open and closed voids be linked with the membrane properties? Line 298, this paragraph shows that “the concentration of TMC monomer is shown to serve as a handle to tune the morphology”. The authors should link the void space percentage of the PA membrane (maybe the freestanding PA membranes) with their physical properties (such as density and mechanical properties) and solvent permeance.

Reply: We thank the reviewer for helping us clarify the effect of open and closed voids on the permeance and mechanical properties of PA membranes. In summary, the open and closed voids would impact the permeance differently, which we took into account in our permeance calculation. For the effective modulus of the membrane, given the experimental data we have, they depend on the fraction of total voids over the membrane, not clearly on the void type. In the revised manuscript, we made the following revisions.

1. The permeance model 2 presented in the original manuscript has already incorporated the contribution of both open and closed void surface areas as separate terms. To make this point more clear, we added the following revision to the main text, and the “Methods” section:
“In our model, we assume that the open voids increase the surface area for solvent transport, whereas closed voids created additional barriers for solvent transport.”
“Approach 1 estimates the membrane nominal thickness using open and closed void surface area, A_{ov} and A_{cv} , by assuming that open void surface areas are surrounded by t_1 thicknesses, the closed void surface areas are enclosed by one layer of t_2 thicknesses and another layer of t_3 thicknesses. The flat membrane region is of t_2 thicknesses. All the assumptions are from direct observations of tomographic reconstructions.”
2. We revised Supplementary Fig. 13 to incorporate a schematic of the local thickness associated with open and closed voids. In addition to the tabulated open and closed void areas (Supplementary Table 5) in the original manuscript, we also incorporated a plot of the percentage surface areas of open and closed voids for the three PA membranes under study.

Supplementary Fig. 13. Void maps coloured to show separate individual void islands. (A) Void island regions for a second tomographic reconstruction of PA1, PA2 and PA3 in addition to those shown in Fig. 3d. (B) Individual void volume distributions for maps shown in (A). (C) Percentage surface areas of open and closed voids. The open and closed void surface area values are tabulated in Supplementary Table 5. (D) Schematic of thickness values observed from tomographic imaging used as assumptions in permeance modelling. It is assumed that open void surface areas are surrounded by t_1 thickness, and the closed void surface areas are enclosed by one layer of t_2 thickness and another layer of t_3 thickness at the base. The flat membrane region is of t_2 thicknesses. All assumptions were based on observations from 3D reconstructions.

- To explain the effect of voids on the mechanical properties of PA membranes, we added the following revision to the main text. Note that here what seems more important is the volume ratio of voids to materials; the effect of open versus closed voids is not clear.

“Furthermore, considering the nanomorphology descriptors presented in this work, void parameters show a large variation among the three membranes (Fig. 4f, Supplementary Tables 4,5), whereas the local thicknesses all fall within the range of 2.5–25 nm (Figs. 2c). Therefore, voids play the greatest role in trending the nanoscale apparent modulus. Previous work theoretically calculated the Young’s modulus for low dielectric constant porous materials, to demonstrate that as porosity decreases the Young’s modulus increases (*Langmuir* **36**, 9377–9387 (2020)). To investigate the potential effect of void parameters in the modulus of the three PA membranes, we define the material fraction (f_{material}) and void fraction (f_{void}) as shown in equations (1) and (2).

$$f_{\text{material}} = \frac{v_{\text{material}}}{(v_{\text{material}} + v_{\text{void}})} \% \quad (1)$$

$$f_{\text{void}} = \frac{v_{\text{void}}}{(v_{\text{material}} + v_{\text{void}})} \% \quad (2)$$

PA1 has the highest f_{void} and lowest f_{material} , with f_{material} increasing and f_{void} decreasing from PA1 to PA3, where f_{material} and f_{void} values are respectively, PA1= $79.5 \pm 0.2\%$ and $20.5 \pm 0.2\%$, PA2= $83.0 \pm 0.4\%$ and $17.0 \pm 0.4\%$, and PA3= $84.0 \pm 0.8\%$ and $16.0 \pm 0.8\%$ (Supplementary Table 4). Considering the effect of voids alone on the modulus, PA1 with the highest void volume, should be easier to indent with the AFM cantilever. Thus, based on void parameters alone, the modulus is

expected to adopt the trend of PA1 < PA2 < PA3, corroborating the GT-based analysis of skeleton rigidity with the mechanical properties of PA membranes.”

Comment 5: *The authors characterize the membrane mechanical strength by AFM. Have the authors tried to characterize the freestanding PA membrane by DMA?*

Reply: We would like to note that without the PS support, freestanding PA membranes of 10–400 nm in thickness are too fragile to be measured by DMA. In addition, the mechanical property of PA membranes matters for their usage under hydraulic pressure, not about their strength when stretched horizontally. That is the reason why in the literature on PA membranes, nanomechanical properties were measured using AFM or related methods as shown in *Sci. Adv.* **8**, eabk1888 (2022) and *Polymer* **255**, 125167 (2022). There are no previous studies using DMA characterization for PA membranes of nanoscale thicknesses. Instead, bulk mechanical properties of PA thin films are characterized by observing the wrinkles of the PA membrane on an elongated and then released sheet of PDMS, followed by observation under AFM (*Science* **348**, 1347–1351 (2015), *J. Membr. Sci.* **574**, 1–9 (2019)) which is beyond the scope of our study.

We would like to emphasize that the focus of our studies was to characterize the nanomechanical properties the membranes. Nevertheless, we tried to characterize the freestanding membranes using DMA and the freestanding membranes broke even when attaching to the sample holder of DMA, as the PA membranes under study are thin films with thicknesses of ~400 nm.

1. To address this comment, we have made the following changes to emphasize that the mechanical properties under study are nanomechanical properties:

“GT representations of PA membranes with interconnected nanovoids relate to their apparent nanomechanical moduli.”

“Using this multifaceted approach, we present for the first time the relationship of nanovoids with membrane morphogenesis, methanol separation performance and nanomechanical properties in the form of the apparent modulus for PA membranes with nanostructures similar to those used in industrial applications.”

2. Added the bulk moduli value ranges of PA membranes from previous work as comparison for the nanomechanical apparent modulus we measured:

“However, the mean apparent moduli show an increasing trend as c_{TMC} increases from 0.05 to 1 w/v% (from PA1 to PA3) with values of 1.12 ± 1.38 GPa, 1.51 ± 1.61 GPa, 2.50 ± 1.73 GPa, respectively (Fig. 5i). These values are on the same order of magnitude as bulk measurements for PA membranes (ranging from 0.1–3.6 GPa) and our recent published work (*Sci. Adv.* **8**, eabk1888 (2022), *Science* **348**, 1347–1351 (2015), *J. Membr. Sci.* **574**, 1–9 (2019)).”

Comment 6: *And Fig. 4h shows the apparent moduli increases as c_{TMC} increases, which is in an opposite trend of the degree of crosslinking (Supplementary Fig. 10b). The author may justify this. Can this trend be linked with the nanomorphology variation?*

Reply: We thank the reviewer for this great point. As noted in the original manuscript, the modulus measured by AFM is an apparent modulus, which depends on both material properties (such as degree of crosslinking) and nanomorphology (such as voids and local thicknesses). As seen in Fig. 2c, the local thicknesses of the membranes range from 2.5–25 nm for all three membranes. Hence, the influence of local thickness on the apparent modulus values is comparable in the three PA membranes under study. It is our understanding that voids play the greatest role in trending the nanoscale apparent modulus.

We revised the main text by adding discussions and equations to explain the effect of voids on the mechanical properties of PA membranes, as addressed in detail in our response to Comment 4, Reviewer 2.

Responses to Reviewer 3

Reviewer 3 commented that “*The study uses novel approaches, provides novel insights, is interesting, and I believe will ultimately be an important addition to the literature on polyamide membranes.*” We thank the reviewer for the insightful, detailed and thorough comments, all of which we have addressed with simulations (new Fig. 3, Supplementary Fig. 10) and discussions (Supplementary Note 4).

Comment 1: *One point that stood out to me was that this being a paper about nanovoids in interfacially polymerized polyamide films, there was no discussion on how the observations by the authors validate or are in conflict with the prevalent theory on how these nanovoids originate. There is relatively extensive experimental work demonstrating that nanobubbles created during the reaction of MPD and TMC give origin to the nanovoids. Much of this work has been led by the Tang laboratory in Hong Kong University (e.g., <https://pubs.acs.org/doi/10.1021/acs.est.9b05892>). Others have used this theory to rationalize novel polyamide film structures (e.g., very large voids, multi-layered void structures) obtained with IP conditions different from typical conditions (<https://doi.org/10.1016/j.memsci.2021.120112>). Given that the authors have performed novel analysis of physical and in silico interfacially polymerized polyamide films, do the authors find their results consistent with this prevalent theory about nanobubble induced void formation? Tang et al demonstrated that by eliminating nanobubble formation, they also eliminated nanovoids. However, the authors do not account for bubble formation in their coarse grain polymerization model and still seemingly obtain voids in the polymer structure? How do the authors reconcile these two results? Overall, a discussion of how the submitted work is consistent or not (and why) with the nanobubble theory of void formation is important.*

Reply: We thank the reviewer for highlighting the importance of discussing literature on void formation. The formation of voids—i.e. the crumpled nanomorphology—of PA membranes was thoroughly discussed in previous studies with debating origins, including nanobubble formation (*Environ. Sci. Technol.* **54**, 3559–3569 (2020)), interfacial heat release (*Science* **348**, 1347–1351 (2015), *Sci. Adv.* **8**, eabm4149 (2022)), and the intrinsic formation process (*J. of Membr. Sci.* **527**, 180–190 (2017), *Sci. Adv.* **8**, eabk1888 (2022), *J. Membr. Sci.* **475**, 504–510 (2015)), among others. In our original manuscript, we discussed some of these processes (e.g., interfacial boiling, rates of monomer diffusion, local temperature fluctuations) as hypotheses from previous work for mechanisms of nanomorphogenesis of the PA membranes. Our experiments show a good agreement with the reaction-diffusion mechanism.

In our CGMD simulations, our primary focus is on investigating the mechanism of monomer diffusion, oligomer coalescence, and membrane growth. The void formation observed in these CGMD simulations arises from the intrinsic growth process. From our previous work on PA membranes (*Sci. Adv.* **8**, eabk1888 (2022)), our current observations from CGMD simulations and our experimental finding from electron tomography, we arrived at an “oligomer-coalescence and growth” mechanism of morphology (local thickness and void) formation which matches with reaction-diffusion instability driven processes.

The different conclusions for the mechanism of morphology formation between the works of Tang and co-workers and our work may simply be due to the different methods of membrane synthesis. The former uses a method of soaking a substrate in a solution of amine monomer, followed by removing the excess and reacting with a solution of acyl chloride monomers. The cycle is repeated to form a multilayer membrane with micrometer thickness (*J. of Membr. Sci.* **540**, 10–18 (2017)). On the contrary, our PA membranes were synthesized using a sacrificial membrane on the substrate to “wet” the substrate with amine monomer. After reaction, our method results in a single layer of PA membrane with nanometer thickness. However, it is important to note that our simulations may not fully capture the mechanism of void formation from nanobubbling, as a result of (i) the simulation size being much smaller than the experimental system ($100 \times 130 \times 100$) nm³ and (ii) the use of an implicit solvent, both of which are strategies to overcome challenges in computation time. Our findings from CGMD simulations on void formation are elucidated in detail in our response to Comment 2, Reviewer 1.

To address this comment, we have made the following revisions:

1. Revised the main text to discuss nanobubble formation as a mechanism of nanovoid and crumple formation for PA membranes:

“One such hypothesis is that during IP, PA first forms a flat membrane and blocks the monomers from reacting further, acting as a self-limiting barrier (*J. Membr. Sci.* **475**, 504–510 (2015)). The subsequent protrusion, void and crumple formation has debated origins ranging from interfacial boiling (*J. Membr. Sci.* **566**, 329–335 (2018)), different rates of monomer diffusion (*J. Membr. Sci.* **475**, 504–510 (2015)), or increases in local temperature (*Science* **348**, 1347–1351 (2015)). In addition, Tang and co-workers propose that nanovoids and crumples form as a result of nanobubbling of carbon dioxide from the heat and hydrogen ions generated during IP (*Environ. Sci. Technol.* **54**, 3559–3569 (2020), *J. of Membr. Sci.* **582**, 342–349 (2019)).”

2. Added discussion to address the inconsistencies between different hypotheses of membrane formation from previous work and clarify the nanovoid and crumple formation hypothesis proposed in this work:

“These hypotheses are directly linked to the high rate of product formation and IP reaction with characteristic dimensions at the nanoscale, which is challenging to probe experimentally (*J. Membr. Sci.* **656**, 120593 (2022)). Hence, the different observations and conclusions of morphology formation might be a result of different methods of synthesis and simulation conditions.”

“Overall, our CGMD simulations along with our experimental observations of multimodal local thickness increments support the mechanism of oligomer-coalescence and growth process of morphogenesis of interconnected PA membranes, supporting that the key morphological features of the PA membrane (i.e. voids and local thickness) are determined by reaction-diffusion instabilities.”

3. Added new discussion and figures (Fig. 3d and Supplementary Fig. 10) to explain void formation in CGMD simulations. Please refer to our response to Comment 2, Reviewer 1.

Comment 2: *Another point that stood out to me is that throughout the paper the authors do not compare any of the values they obtain from the analysis of their physical or in silico membranes (i.e., permeance, thickness, void fraction, etc.) or membrane fabrication values (i.e., MPD and TMC concentrations and ratios) with those used typically for production of well-performing polyamide membranes in the industry or literature. Such comparisons are important to put in perspective the results and conclusions reported by the authors. For example, were the concentrations/ratios used to prepare the membranes consistent with what is typical for well-performing membranes? If not, please specify why the concentrations/ratios used in this study were chosen? How does the permeance, thickness, void fraction, crosslinking, etc., of the physical and in silico membranes compare with values in the literature for commercial or lab-made membranes?*

Reply: We thank the reviewer for the great comment! In the revised manuscript, we added discussions on comparisons of our findings with previous literature to address the comment as follows.

1. Added comparison of the monomer concentrations with previous literature in the main text:

“For the three samples of our focus, the concentration of the MPD monomer (c_{MPD}) and the reaction time are maintained constant at 5 w/v% and 60 s, respectively, whereas the concentration of the TMC monomer (c_{TMC}) is varied from 0.05 (PA1), 0.1 (PA2) and 1 w/v% (PA3) (Fig. 1b, Supplementary Fig. 1, Supplementary Movie 1). Our previous work showed that high c_{MPD} results in PA membranes with interconnected crumples, similar to those used in commercial applications (*Mol. Syst. Des. Eng.* **5**, 102–109 (2020)). Previous work on polyamide thin-film membranes used comparable monomer concentrations and reported methanol permeance values more than 20 times as those of commercial thin-film membranes (*Science* **348**, 1347–1351 (2015)).”

2. Added in the main text discussions on the local thickness, void volume, degree of crosslinking and by employing this multi-scale modelling technique permeance of our PA membranes to the values previously reported:

“Previous studies of PA membranes measured average thickness values of 14–30 nm using cross-sectional scanning electron microscopy (SEM) (*J. Membr. Sci.* **475**, 504–510 (2015)), which align with t_2 and t_3 local thickness values reported here. However, unlike previous work, the tomography-based method we adopt is capable of mapping the local thickness of less abundant thin-walled membrane regions (t_1 , Fig. 2c), showing local thickness values of 5–10 nm.”

“The void spaces of the three membranes contribute to 20.5 ± 0.2 %, 17.0 ± 0.4 % and 16.0 ± 0.8 % of the total volume in PA1, PA2 and PA3 respectively (Supplementary Table 3), within the range of the void fractions (15% to 35%) previously reported for PA membranes using spectroscopic methods and microscopic characterization (*J. Membr. Sci.* **497**, 365–376 (2016), *J. Membr. Sci.* **500**, 124–135 (2016), *Ind. & Eng. Chem.* **60**, 2898–2910 (2021)).”

“The degree of crosslinking of the membranes determined by X-ray photoelectron spectroscopy (XPS) (Supplementary Fig. 7, Supplementary Table 3), is 97.7 %, 73.9 % and 41.8 % for PA1, PA2 and PA3, respectively, similar to values reported in previous literature ranging from 42%–99% (*Science* **348**, 1347–1351 (2015), *Sci. Rep.* **6**, 22069 (2016)).”

“It should be noted that the experimentally measured methanol permeance of the PA membranes in this study is within the same order of magnitude of similar membranes studied in previous work (*Sci. Adv.* **8**, eabk1888 (2022), *Science* **348**, 1347–1351 (2015)).”

Comment 3: *The authors say vaguely (pages 15-16) how they 'predicted' solvent permeance; they indicated they used the Spiegler-Kedem model or some modification of it. However, they do not explain in the Results, Methods, or Supplementary Materials what transport equations they used for this, what the inputs were to the equations, etc. They only provide equations about thickness (eqs 2, 3) and other basic coefficient equations about coefficients in transport models (eqs 4-6). These, however, are not sufficient to understand how the transport was modeled. For example, the authors state that they used the degree of crosslinking in the modeling but none of the equations presented have crosslinking as one of the parameters. It is important the authors explain the transport modeling clearly (even if in the Supplementary Materials) since results from that modeling are an important part of Figure 3 and corresponding discussion.*

Reply: We thank the reviewer for giving us the opportunity to make the paper stronger! In the original manuscript, we referred to our previous work (*Sci. Adv.* **8**, eabk1888 (2022)) and skipped the equations and derivations that were included in that paper. We realized that indeed this has caused confusion and have now added a new Supplementary Note 4 to include the complete details on the derivation and assumptions used for the model relating membrane morphology to methanol permeance. We used the Spiegler-Kedem model as the transport equation (equation (1), (2) in Supplementary Note 4). Specifically, we used the surface areas of the membrane, void surface areas, local membrane thickness, thickness from AFM and degree of crosslinking as inputs, all of which are listed in Supplementary Tables 2–5.

The reviewer also notes that it is unclear how the degree of crosslinking was incorporated into the model, which is an important feature of our work. To further elaborate, the degree of crosslinking was found to affect the polymer film density, which is a property that may affect fluid permeability, and therefore we used the modified Spiegler-Kedem model from our previous work (*Sci. Adv.* **8**, eabk1888 (2022)) that takes membrane density into account, shown in equation 12 of Supplementary Note 4. Based on previous work and literature values (*Science* **348**, 1347-1351 (2015), *Sci. Adv.* **8**, eabk1888 (2022)), density is a linear function of the degree of crosslinking within our range of interest ($41\% < \text{DOC} < 98\%$) with an R^2 of 0.98. With this relationship, we accounted for the degree of crosslinking within the fitting.

In the revised manuscript, the newly added Supplementary Note 4 is as follows.

Supplementary Note 4: Derivation of solvent permeance fittings

“We used the Spiegler-Kedem model as the transport equation for the permeance fitting as shown in equation (1) below, where J_v is the volumetric flux, L_p is the hydraulic permeance, ΔP is the transmembrane pressure, σ is the reflection coefficient and $\Delta\pi$ is the osmotic pressure.

$$J_v = L_p(\Delta P - \sigma\Delta\pi) \quad (1)$$

In the Spiegler-Kedem model, L_p is given by the solvent-membrane permeability (P_m) and the thickness of the membrane (δ), which is used as the basis to derive our fitting (equation (2)).

$$L_p = \frac{P_m}{\delta} \quad (2)$$

To determine the nominal membrane thickness, the membrane was categorized into three distinct regions, which are depicted in Supplementary Fig. 13. Region 1 is the thin open void, region 2 is the featureless base layer, and region 3 is the thick closed void section.

Tomographic reconstructions were used to calculate the top (A_T) and bottom (A_B) surface areas of the PA membrane, the surface area of open voids (A_{OV}) and the surface area of closed voids (A_{CV}). The geometric area (A_G) is the projection area of the tomographic reconstruction (Supplementary Table 5).

Two approaches are used to fit the experimental methanol permeance. The first fitting (Approach 1) used open and closed void areas, and the local membrane thicknesses measured using tomographic reconstruction.

For Approach 1, the surface areas of region 1 and region 3 are given by A_{OV} and A_{CV} respectively, and the surface area of the featureless region 2 (A_F) is given by equation (3).

$$A_F = A_T - (A_{OV} + A_{CV}) \quad (3)$$

Using the surface area data from tomography, the area percentage for each region as shown in equations (4), (5) and (6) where region 1, region 2 and region 3 are represented by $\%A_{open}$, $\%A_{flat}$ and $\%A_{closed}$, respectively.

$$\%A_{open} = \frac{A_{OV}}{A_{OV} + A_{CV} + A_F} \quad (4)$$

$$\%A_{flat} = \frac{A_F}{A_{OV} + A_{CV} + A_F} \quad (5)$$

$$\%A_{closed} = \frac{A_{CV}}{A_{OV} + A_{CV} + A_F} \quad (6)$$

Three assumptions are driven by observations from the 3D tomographic reconstructions of PA membranes, thickness mapping and void reconstructions; (i) open void regions are surrounded by t_1 and t_2 thickness, (ii) closed void regions are enclosed by one layer of t_2 thickness and another layer of t_3 thickness at the base, and (iii) the flat membrane region is of t_2 thicknesses.

Using the above assumptions, the nominal open void thickness (region 1), δ_{open} , was estimated using the multimodal thickness maxima from tomography (t_1 , t_2 , t_3) and their weighted percentages (x_1 , x_2 , x_3) as shown in equation (7).

$$\delta_{open} = \frac{x_1}{x_1 + x_2} * t_1 + \frac{x_2}{x_1 + x_2} * t_2 \quad (7)$$

To estimate the nominal closed void thickness (region 3), δ_{closed} , the thickness maxima enclosing the top void region, t_3 , was added to the thickness maxima representing the bottom surface thickness t_2 (equation (8)).

$$\delta_{closed} = t_2 + t_3 \quad (8)$$

The flat featureless layer (region 2) was estimated with the thickness maxima t_2 (equation (9)).

$$\delta_{bottom} = t_2 \quad (9)$$

To calculate the nominal thickness ($\delta_{nom,1}$) from these three distinct regions, a harmonic mean of each region's thickness weighted by the percentage of area of that region was used as shown in equation (10).

$$\delta_{nom,1} = \left(\frac{\%A_{open}}{\delta_{open}} + \frac{\%A_{closed}}{\delta_{closed}} + \frac{\%A_{bottom}}{\delta_{bottom}} \right)^{-1} \quad (10)$$

The second fitting (Approach 2) used membrane thicknesses derived from AFM measurements ($\delta_{nom,2}$) as shown in Supplementary Fig. 14.

For both approaches, the permeability (P_m) was then calculated given the nominal membrane thickness and experimentally determined solvent permeance ($L_{exp,k}$) with the following equation:

$$P_{m,k} = L_{exp,k} \times \delta_{nom,k} \quad (11)$$

The material properties of the membranes were assumed to be dependent on the mass. A permeability constant (μ) was defined to isolate the effect of polymer density (equation (12)), as the solvent-membrane permeability is inversely proportional to membrane density. Note that the density is a linear function of the degree of crosslinking within our range of interest (41% < DOC < 98%) with an R^2 of 0.98, thus the density was calculated using DOC based on previous work (*Science* **348**, 1347-1351 (2015), *Sci. Adv.* **8**, eabk1888 (2022)).

$$\mu_k = P_{m,k} \times \rho_k \quad (12)$$

The permeability constant for each PA membrane was averaged to form an average permeability constant (μ_{avg}), and used to calculate the solvent permeance ($L_{fit,k}$) as follows (equation (13)).

$$L_{fit,k} = \frac{\mu_{avg}}{\rho_k \times \delta_{nom,k}} \quad (13)$$

Comment 4: *It is unclear what the purpose of the comparison between ‘model estimates’ and experimental measures of permeance is. The authors use the word “predict/prediction” for the model estimates but the “permeability” constant used in the model estimates of permeance (i.e., mu in eqs 5, 6) was calculated directly from the experimental data (permeability, thickness, and density). To calculate the model estimate, the authors simply used the same equation (eq 6) used to analyze the experimental data (and the permeability extracted from it using the experimental data) but now with the densities and thicknesses obtained from the polyamide layers generated in silico. Therefore, the model estimates are not a true prediction of permeance (i.e., because the calculations used the permeability obtained from experiment). The only purpose I can see for comparing ‘model estimates’ and experimental measures of permeance is to evaluate whether the product of thickness and density for one case (e.g., in silico) is consistent or not with the other case (e.g., experiment).*

Reply: Thank you! We realized the confusion was again caused by us omitting the equations referred to in the earlier work. All the model discussions relating the membrane nanomorphology to permeance were based on experimental data, and involve no data (e.g., chemical properties, morphology) from simulations. We would like to emphasize “permeability” and “permeance” are two different concepts. Permeability is a

materials-specific property, much like density, which is invariant of the materials' nanomorphology. For permeability, we obtained the values from the fitting of different membrane samples. Permeance, on the other hand, depends on the nanomorphology, in particular, the local thickness and voids.

The reviewer makes great comments about the permeance model regarding its purpose, and our response to the reviewer's previous comment (comment 3) will better illuminate the assumptions in our model. For full clarity, the model's purpose was to connect morphological data to the experimental permeance results, if possible, and the morphological data we collected was over-defined, i.e., the data set allowed multiple ways to model the membrane permeance. Therefore, multiple models, using data from different morphological techniques, were constructed to form an "apples to apples" comparison of the quality of fit for models of various morphological techniques. Our conclusion was that the TEM-derived method for estimating the open and closed void areas, developed for the first time in this work was the best at matching experimental permeance trends.

For literature references to the fitting procedure, Spiegler and Kedem were the first to build the foundational membrane permeance model in 1966, referred to as the Spiegler-Kedem model (*Desalination* 1, 311–326 (1966)). The permittivity constant of the Spiegler-Kedem model was further defined to account for density changes by H. An *et al.* in 2022 (*Sci. Adv.* 8, eabk1888 (2022)), and this is the basis for our permeance fit.

To further address this comment, we revised the terminology of the main text and Supplementary Note 4. In the revised manuscript, the term "model" is replaced by "approach" and "prediction" is changed to "fitting".

Comment 5: *The definition of local thickness (L159-160) is not very clear. Does 'local thickness' refer to that of a film of continuous polyamide or to a descriptive thickness at a specific position in the x-y plane? For example, in Figure 3b(ii) it is clear that in some instances when moving along the z-axis (i.e., perpendicular to the x-y plane) one traverses polyamide, then void, then polyamide again (e.g., right hand side of Fig 3b(ii)). Would there be two local thicknesses at that x,y location? Or just one? If one, what would be the thickness at that x,y location? The summation of the diameter of the sphere that fits into the top polyamide film plus the diameter of the sphere that fits into the bottom polyamide film? Please clarify. My understanding is that the thickness mappings in figures and videos are associated with x,y locations.*

Reply: Thank you! The local thickness is measured at a (x, y, z) position containing the PA membrane, or per voxel in the reconstructed 3D tomograph. As discussed in the original manuscript, "Here, the local thickness of the membrane is defined as the diameter of the largest sphere inscribed within the membrane containing a given voxel (*Microsc. Microanal.* 13, 1678–1679 (2007))". Based on this definition, we can have a thickness value at each voxel, essentially as a matrix in 3D. The sliced view is to show at each fixed z' coordinates, what are the corresponding thickness value at the locations of (x, y, z') . For (x,y,z) locations belonging to the void region without materials, the local thickness is no longer relevant and is assigned to zero. The 3D volume is essentially converted into a thickness map of 3D spheres. Supplementary Movie 2 shows the 3D volume filling with spheres).

To address this comment, we revised Fig. 2a and the caption. The schematic now shows only a few spheres/circles in the left panel to show that the spheres capture the volume. Coordinate arrows were added for clarification.

Fig. 2. (a) Schematic showing volume reconstruction cross-section with inscribed spheres used to measure local thickness and sliced views of local thickness maps. The tomographically reconstructed 3D volume is filled with spheres (left) to generate a 3D thickness map (middle). The thickness map can be viewed as slices along the height (right) to observe local thickness changes and voids.

Comment 6: I am not convinced the important statement in L172-L175 that $t_2 \sim 2t_1$ and $t_3 \sim 3t_1$ is accurate. From Fig 2 it appears that $t_1 \sim 5\text{nm}$, $t_2 \sim 15\text{nm}$ and $t_3 \sim 20\text{nm}$. Therefore $t_2 \sim 3t_1$ and $t_3 \sim 4t_1$. This is an important detail because then the reasoning for the origin of the thicker regions may require some revision. Can the authors tabulate these thicknesses?

Reply: Thank you! We tabulated the thickness values in the original manuscript as Supplementary Table 2.

Supplementary Table 2. Thickness maxima values of the multimodal distribution of local thickness histograms for PA1, PA2 and PA3.

Thickness maxima	PA1	PA2	PA3
t_1 (nm)	7.2 ± 0.5	6.9 ± 0.3	5.7 ± 0.3
t_2 (nm)	17.4 ± 0.1	16.4 ± 0.1	14.4 ± 0.1
t_3 (nm)	23.0 ± 0.1	21.9 ± 0.2	20.4 ± 0.1

As seen here, apart from slight deviations, all the thickness values follow the $t_2 \approx 2t_1$ and $t_3 \approx 3t_1$ observations. We reason that slight deviations are a result of comparing only the maxima of Gaussian-fitted broad multimodal peaks (Supplementary Fig. 6).

To address this comment, in the revised manuscript, we added the thickness values to the main text.

“Moreover, as shown in Fig. 2e, the second (t_2) and third (t_3) local thickness maxima are approximately twice and three times the value of the first (t_1), supporting that two or more crumple walls connect or merge to form thicker local regions ($t_2 \approx 2t_1$ and $t_3 \approx 3t_1$, where $t_1 = 7.2 \pm 0.5$, 6.9 ± 0.3 and 5.7 ± 0.3 nm, $t_2 = 17.4 \pm 0.1$, 16.4 ± 0.1 , 14.4 ± 0.1 nm, $t_3 = 23.0 \pm 0.1$, 21.9 ± 0.2 , 20.4 ± 0.1 nm respectively, for PA1, PA2 and PA3; Supplementary Table 2).”

Comment 7: The main use of polyamide membranes in real life (and in academic literature) is water treatment. Therefore it seems strange that the authors evaluated methanol permeance instead of water permeance for this study. I am not aware of the use of polyamide membranes for methanol filtration in industry as an established practice (though perhaps I may just be unaware of it). The paper would be stronger if it evaluated water permeance, though if that is not possible at the moment, providing a reason to evaluate methanol permeance instead of water permeance may be informative to the reader.

Reply: We thank the reviewer for the great comment. We have included more discussions on organic solvent nanofiltration using polyamide membranes in the revised manuscript.

“This level of detail in PA membrane void reconstruction paves the way for better understanding the effect of nanoscale void morphology on solvent permeance. PA membranes are employed in pharmaceutical, chemical and petrochemical industries to separate and recover organic solvents used as raw materials or cleaning agents in a process called organic solvent nanofiltration, which

can separate organic impurities in the range of 200–1000 Da from organic solvents ((*Science* **348**, 1347–1351 (2015), *J. of Membr. Sci.* **647**, 120306 (2022)). The use of polymer films, such as PA membranes for organic solvent nanofiltration is promising because it is an energy-efficient and cost-effective process (*J. of Membr. Sci.* **601**, 117932 (2020)). Therefore, to better understand the separation behaviour of organic solvents, we explore the permeance of methanol across the three PA membranes under study.”

Other comments

L152-L154: it is unclear what the reference to Geise’s work is for. Do the authors mean that Geise et al did not have access to accurate thickness values and therefore their prediction of the upper bound could be improved in the future with more accurate thickness measurements like the ones in the submitted work? Or did the authors simply mean to give an example of when accurate thickness values may be needed?

Reply: We thank the reviewer for the comment. The main text is revised to address the comment as follows.

“Geise et al. (*J. Membr. Sci.* **369**, 130–138 (2011)) demonstrated an example of where accurate membrane thickness values are needed by using solution–diffusion theory, which requires knowledge of the thickness of the active layer, to model the upper bound of the inverse correlation between water permeability and salt selectivity in PA thin-film desalination membranes.”

L218-L220: I do not understand the mechanism the authors try to explain in one sentence here. They mention “this instability” but they do not mention any instability in the preceding sentences.

Reply: This is a typographic error. We thank the reviewer for the comment and we added the “Turing diffusion-reaction Instability” into the main text.

“Our previous work on PA membranes of spatially separated crumples showed that a Turing diffusion-reaction instability accounts for many of the characteristics of PA membranes prepared by IP: (i) the protrusion of the IP reaction front ultimately into crumples and (ii) more quantitatively, their average lateral separation, or “wavelength”, which follows a power law dependence on the monomer concentrations. We proposed that the final film morphology corresponds to the point at which the Turing instability of the reaction front is “frozen” by the eventual coalescence of a contiguous membrane that blocks further diffusion of MPD into the hexane phase.”

The statement in L317 “that monomer concentration alone is also a good strategy to increase void volume” is supported by the results in a recent relevant study (<https://doi.org/10.1016/j.memsci.2021.120112>) that demonstrated experimentally that the MPD concentration and availability affects polyamide film thickness and void structure and fraction.

Reply: Thank you! The paper shared by the reviewer was cited in the main text.

“As previous work has demonstrated (*J. Membr. Sci.* **644**, 120112 (2022)), we show that monomer concentration alone is also a good strategy to increase void volume.”

Fig 1a: The authors state in the caption that “The schematic is zoomed-in to show the interconnected crumple morphology” but that interconnected crumple morphology is not clear. Making bigger the inset may suffice.

Reply: Thank you! The inset was made larger in Fig. 1a as follows. The caption was also revised.

Fig. 1. (a) IP of MPD in aqueous phase and TMC in organic phase on the sacrificial and porous support layers. The schematic is zoomed-in (black box and dotted lines) to show the interconnected crumple morphology.

Comment 5: Fig 2a: the left panel (especially what the spheres correspond to) is not very clear.

Reply: Thank you for the opportunity for clarification! Fig. 2a was revised to show that the spheres in the left panel corresponds to the thicknesses in the middle panel. Please refer to Comment 5, Reviewer 3 for the revised figure.

Fig. 3d: what do the colors represent?

Reply: Thank you! The void islands were assigned random colours to make the individual voids clear and separate them. The figure caption was revised to clarify this:

“Void maps are assigned colours randomly to clearly show separate individual void islands.”

Equations 2 and 3: what is the conceptual basis for these equations. Please add to the Supplementary Materials.

Reply: We thank the reviewer for their comment. We added a new Supplementary Note 2 with detailed derivations and discussions. Please refer to Comment 3, Reviewer 3.

Supplementary Fig 4c: it is unclear what this figure shows, especially the right panel of the image.

Reply: Thank you! The figure was revised as follows:

Supplementary Fig. 5. Tomographic reconstruction shows complex networked crumple morphology of PA membranes. TEM micrograph (A) and 3D tomographic reconstruction (B) of PA membrane. (C) Two volume slices at different z -heights of the 3D reconstruction in (B) showing the internal nanomorphology with voids and varying local thickness, which is not captured in (A).

REVIEWERS' COMMENTS

Reviewer #1 (Remarks to the Author):

The authors have addressed my comments and this is a very nice contribution worthy of publication in Nature Comm.

Reviewer #2 (Remarks to the Author):

This resubmitted manuscript has undergone significant improvements based on the reviewers' comments, and the authors have provided detailed responses to each comment. Therefore, it is now suitable for acceptance.

Reviewer #3 (Remarks to the Author):

The authors have addressed my comments satisfactorily.